# Connectomic analysis of the *Drosophila* lateral neuron clock cells reveals the synaptic basis of functional pacemaker classes

Orie T Shafer[1†], Gabrielle J Gutierrez[2†], Kimberly Li[3], Amber Mildenhall[3], Daphna Spira[2,3], Jonathan Marty[4], Aurel A Lazar[4], Maria de la Paz Fernandez[3]*

[1]Advanced Science Research Center, The City University of New York, New York, United States; [2]Center for Theoretical Neuroscience, Zuckerman Institute, Columbia University, New York, United States; [3]Department of Neuroscience and Behavior, Barnard College, New York, United States; [4]Department of Electrical Engineering, Columbia University, New York, United States

**Abstract** The circadian clock orchestrates daily changes in physiology and behavior to ensure internal temporal order and optimal timing across the day. In animals, a central brain clock coordinates circadian rhythms throughout the body and is characterized by a remarkable robustness that depends on synaptic connections between constituent neurons. The clock neuron network of *Drosophila*, which shares network motifs with clock networks in the mammalian brain yet is built of many fewer neurons, offers a powerful model for understanding the network properties of circadian timekeeping. Here, we report an assessment of synaptic connectivity within a clock network, focusing on the critical lateral neuron (LN) clock neuron classes within the Janelia hemibrain dataset. Our results reveal that previously identified anatomical and functional subclasses of LNs represent distinct connectomic types. Moreover, we identify a small number of non-clock cell subtypes representing highly synaptically coupled nodes within the clock neuron network. This suggests that neurons lacking molecular timekeeping likely play integral roles within the circadian timekeeping network. To our knowledge, this represents the first comprehensive connectomic analysis of a circadian neuronal network.

*For correspondence:
mfernand@barnard.edu

†These authors contributed equally to this work

Competing interest: The authors declare that no competing interests exist.

## Editor's evaluation

This paper analyzes the synaptic connections of two subsets of the *Drosophila* clock neurons, the small ventral lateral neurons, and the dorsal lateral neurons. The study reveals high heterogeneity of the dorsal lateral neurons and the potential involvement of non-clock neurons that might act as inter-clock neurons. The manuscript will be of interest to chronobiologists and neuroscientists working on neuronal networks and it provides new insights into circadian clock network organization that may be of general value. The data analysis is rigorous, and the conclusions are justified by the data.

## Introduction

Most organisms undergo predictable daily changes in gene expression, physiology, and behavior that are driven by endogenous circadian clocks (*Aschoff, 1981*; *Takahashi et al., 2001*). In animals, the molecular feedback loops underlying circadian clocks are present in all organ systems but are required

**eLife digest** Most organisms on Earth possess an internal timekeeping system which ensures that bodily processes such as sleep, wakefulness or digestion take place at the right time. These precise daily rhythms are kept in check by a master clock in the brain. There, thousands of neurons – some of which carrying an internal 'molecular clock' – connect to each other through structures known as synapses. Exactly how the resulting network is organised to support circadian timekeeping remains unclear.

To explore this question, Shafer, Gutierrez et al. focused on fruit flies, as recent efforts have systematically mapped every neuron and synaptic connection in the brain of this model organism. Analysing available data from the hemibrain connectome project at Janelia revealed that that the neurons with the most important timekeeping roles were in fact forming the fewest synapses within the network. In addition, neurons without internal molecular clocks mediated strong synaptic connections between those that did, suggesting that 'clockless' cells still play an integral role in circadian timekeeping.

With this research, Shafer, Gutierrez et al. provide unexpected insights into the organisation of the master body clock. Better understanding the networks that underpin circadian rhythms will help to grasp how and why these are disrupted in obesity, depression and Alzheimer's disease.

within a relatively small population of brain neurons to maintain daily rhythms in behavior and physiology (*Herzog, 2007*). In mammals, the hypothalamic suprachiasmatic nuclei (SCN) are the brain's central circadian clock since the ablation of the SCN or knockouts of the molecular clock within the SCN results in loss of circadian rhythms (reviewed by *Herzog et al., 2017*). Compared to peripheral tissues, timekeeping in the brain's central clock is more resilient to the loss of clock gene function and to environmental perturbations. This resilience comes from physiological connections between SCN neurons (*Buhr et al., 2010*; *Liu et al., 2007*). Determining how clock neurons are connected is, therefore, essential to understand circadian timekeeping in animals.

A fundamental challenge to addressing this question is the complexity of the circadian neuronal networks. The paired mammalian SCN consists of tens of thousands of neurons with diverse neurochemical outputs that form a complex network (reviewed by *Hastings et al., 2019*). The clock neuron network of *Drosophila melanogaster* offers a powerful model to understand the network properties of neural circadian timekeeping. The molecular clocks of insects and mammals are highly conserved and their neuronal networks appear to share common motifs (*Nitabach and Taghert, 2008*). The fly's clock neuron network consists of relatively few neurons that can be organized in a small number of discrete anatomical classes (*Shafer et al., 2006*; *Figure 1A*). Furthermore, genetic tools to specifically manipulate subclasses of the fly circadian network have made it possible to understand the functional roles of these subclasses in the production of endogenous circadian rhythms in sleep and activity (*Ahmad et al., 2021*; *Shafer and Keene, 2021*). Though transgenic markers of synaptic connectivity and live-imaging-based assessments of functional connectivity have been used to address neural connectivity within the clock neuron network (e.g. *Reinhard et al., 2022a*; *Yao et al., 2012*), a systematic assessment of synaptic connectivity within this network is lacking.

The comprehensive mapping of synaptic connectivity using serial electron microscopy and reconstruction of neuronal volumes allows us to assess the network structures of brain circuits (*Kasthuri et al., 2015*). *Scheffer et al., 2020* provided a dense reconstruction of chemical synapses within a large portion of the fly's central brain, called the hemibrain. Annotation of the Janelia hemibrain connectome contains most of the key anatomical subsets of the fly's clock neuron network and powerful computational tools are available to navigate this connectome (*Lazar et al., 2021*; *Scheffer et al., 2020*) making it possible to analyze the structure of the fly's brain clock network.

The *Drosophila* Circadian Neuron Network is a group of ~150 neurons that display synchronous oscillation in PER and TIM abundance and can be subdivided based on gene expression, anatomy, and location in the brain (*Helfrich-Förster, 2005*). These are (i) the lateral neurons (LNs), which can be subdivided into ventral lateral (LNv) and dorsal lateral (LNd) neurons, (ii) the lateral posterior neurons (LPNs), and (iii) the dorsal neurons (DNs), which can be subdivided into dorsal neurons 1, 2, and 3 (DN1, DN2, and DN3) and further subdivided into anterior (DN1a) and posterior (DN1p) DN1s

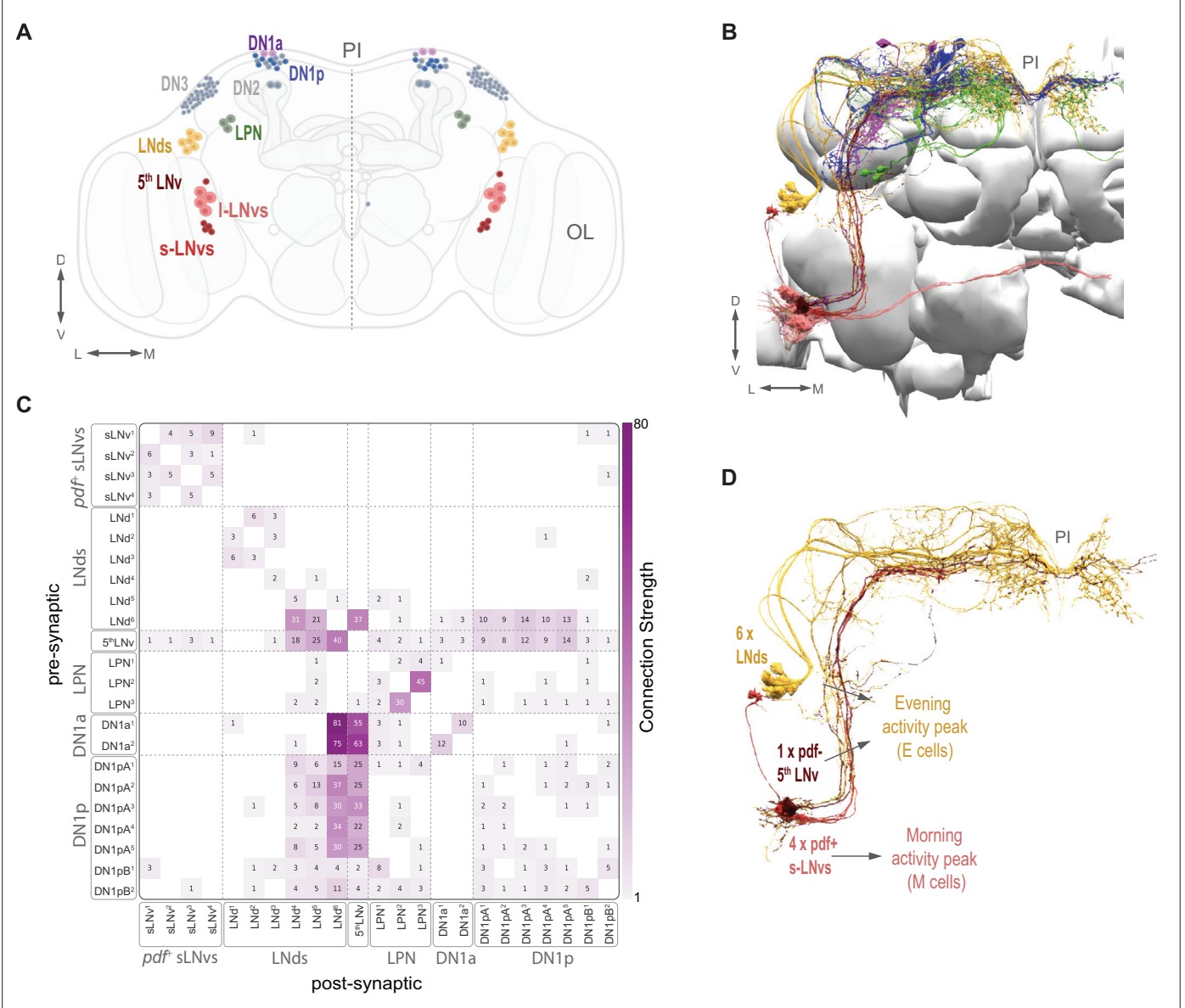

**Figure 1.** The circadian clock neuron network and identified clock neurons in the hemibrain. (**A**) The circadian clock network is shown in both hemispheres. Currently identified cells are shown in color (s-LNvs red, l-LNvs light red, LNds in orange, DN1a in purple, a subset of DN1p in blue) and known clock cells not yet identified within the hemibrain connectome data are shown in gray (DN2, DN3, and a subset of DN1p). The *pars intercerebralis* (PI) and optic lobe (OL) are indicated in gray. (**B**) Identified clock neurons in the hemibrain. Neuropils are shown in gray. Color codes as indicated in (**A**). (**C**) Heatmap indicates synaptic connections strength among all identified circadian clock neurons, including weak (weight <3) connections. (**D**) The timekeeping lateral neurons that compose the morning (**M**) and evening (**E**) Oscillators: s-LNvs (M cells), and 5th LNv and LNds (E cells).

(*Helfrich-Förster et al., 2007b*; Helfrich-Forster, Yoshii, *Helfrich-Förster et al., 2007a*; *Shafer et al., 2006*).

The LNvs release the key circadian neuropeptide Pigment Dispersing Factor (PDF) and can be further subdivided into small and large LNvs (s-LNvs and l-LNvs). Release of PDF from the s-LNvs, which takes place in the dorsal protocerebrum where the LNds and DNs reside, is required for endogenous circadian timekeeping (*Park et al., 2000*; *Renn et al., 1999*). The current annotation of the hemibrain dataset identifies all the expected lateral neuron classes: four l-LNvs, four s-LNvs, the 5th-s-LNv, and six LNds (*Scheffer et al., 2020*; *Figure 1B and C*). However, some clock neurons have not yet been unequivocally identified within the hemibrain dataset: approximately half of the DN1ps (*Reinhard et al., 2022a*; *Scheffer et al., 2020*) and the DN2s and DN3s. The LN classes of clock

neurons represent the minimum sub-network required to produce endogenous bimodal rhythm of sleep and activity. A functional molecular clock only in the s-LNvs, 5th s-LNv, and LNds is sufficient to produce such rhythms (*Grima et al., 2004*; *Rieger et al., 2006*).

Here, we report an assessment of chemical synaptic connectivity for the critical LN clock neuron classes within the hemibrain volume. We have determined the patterns of synaptic connectivity within this core LN circadian sub-network and the connectivity with other clock neurons annotated within the hemibrain dataset. This analysis provides a significant and comprehensive insight into the synaptic circuitry underlying the organization and function of the *Drosophila* central brain clock. Our results reveal that previously identified anatomical and functional subclasses of LNs are distinct connectomic classes that receive unique sources of synaptic inputs and outputs. Furthermore, our analysis reveals a remarkable heterogeneity in how clock neuron subclasses form connections to other clock neurons and identifies a small number of clock cell subtypes that are highly synaptically coupled nodes. Finally, we find 'non-clock neuron' targets which themselves form synapses back onto clock neurons. We propose that those non-clock cells are important components of the timekeeping network.

**Table 1.** Identification of Clock Neurons within the Hemibrain Volume.

| BodyId | Type | Sequential label | Subphase |
|---|---|---|---|
| 2068801704 | s-LNv | s-LNv1 | M |
| 1664980698 | s-LNv | s-LNv2 | M |
| 2007068523 | s-LNv | s-LNv3 | M |
| 1975347348 | s-LNv | s-LNv4 | M |
| 5813056917 | LNd | LNd4 | E1 |
| 5813021192 | LNd | LNd5 | E1 |
| 5813069648 | LNd | LNd6 | E2 |
| 511051477 | 5th LNv | 5th LNv | E2 |
| 296544364 | LNd | LNd1 | E3 |
| 448260940 | LNd | LNd2 | E3 |
| 5813064789 | LNd | LNd3 | E3 |
| 356818551 | LPN | LPN1 | |
| 480029788 | LPN | LPN2 | |
| 450034902 | LPN | LPN3 | |
| 546977514 | LPN | LPN4 | |
| 264083994 | DN1a | DN1a1 | |
| 5813022274 | DN1a | DN1a2 | |
| 5813010153 | DN1pA | DN1pA1 | |
| 324846570 | DN1pA | DN1pA2 | |
| 325529237 | DN1pA | DN1pA3 | |
| 387944118 | DN1pA | DN1pA4 | |
| 387166379 | DN1pA | DN1pA5 | |
| 386834269 | DN1pB | DN1pB1 | |
| 5813071319 | DN1pB | DN1pB2 | |
| 1884625521 | l-LNv | lLNv1 | |
| 2065745704 | l-LNv | lLNv2 | |
| 5813001741 | l-LNv | lLNv3 | |
| 5813026773 | l-LNv | lLNv4 | |

# Results

## The clock neuron network and the hemibrain dataset

Representatives of all classes of clock neurons, with the exception of the DN2 and DN3 classes, have been identified in the hemibrain volume used here (*Figure 1A and B*). See *Table 1* for the naming scheme used here for the clock neurons identified in the dataset with their corresponding unique body IDs. The patterns of connectivity among some of these neurons have recently been briefly described (*Ahmad et al., 2021*; *Reinhard et al., 2022a*; *Reinhard et al., 2022b*). Scheffer and colleagues (*Scheffer et al., 2020*) define synaptic strength within the hemibrain volume by the number of synaptic connections formed between neurons, defining connections consisting of only one or two chemical synapses as weak and subject to error, three to nine synapses as medium strength, and of ten or more synapses as strong. We used these definitions of synaptic strength in this study. As a whole, the identified clock neurons appear to be sparsely interconnected by chemical synapses. Most clock neuron pairs form either no synapses or weak synaptic connections (*Figure 1C*). However, there are a few clear exceptions: a single LNd (LNd6) and the 5th s-LNv form strong connections with one another and with members of the DN1 group (*Figure 1C*; see also *Reinhard et al., 2022a*) and stand out among the clock neurons as hubs of inter-clock synaptic connectivity.

To asses synaptic connectivity in the *Drosophila* clock network, we focused on three critical classes of lateral clock neurons (LNs); the four Pigment Dispersing Factor (PDF) -expressing small ventral lateral neurons (s-LNvs), the six dorsal lateral neurons (LNds) and the PDF negative 5th s-LNv (*Figure 1D*). We have chosen to focus on these clock neuron classes for two reasons. First, these classes are completely accounted for in the hemibrain annotation used here. Second, the 11 neurons that comprise the LN class are sufficient to drive the fly's endogenous bimodal rhythm in locomotor activity (*Grima et al., 2004*; *Rieger et al., 2006*) and therefore represent a critical sub-network within the fly's circadian system.

## The small LNvs are a highly unified connectomic class distinct from the 5th s-LNv

The s-LNvs have long been considered critical circadian pacemakers within the *Drosophila* clock neuron network (*Helfrich-Förster, 1998*). These cells express the neuropeptide PDF (*Helfrich-Förster, 1995*; *Renn et al., 1999*), maintain strong molecular timekeeping under constant conditions (*Shafer et al., 2002*; *Yang and Sehgal, 2001*) and are required for robust endogenous circadian rhythms (*Grima et al., 2004*; *Shafer and Taghert, 2009*). The s-LNvs also contribute to the morning peak of the fly's crepuscular daily activity rhythm (*Grima et al., 2004*; *Stoleru et al., 2004*). In addition to PDF, these cells also release the neuropeptide short Neuropeptide F (sNPF) (*Johard et al., 2009*) and the neurotransmitter glycine (*Frenkel et al., 2017*). Hyper excitation of the s-LNvs leads to a morning-like gene expression profile for many circadian genes, while hyperpolarization leads to an evening-like transcriptional profile (*Mizrak et al., 2012*). Though by no means the only clock neurons capable of producing an endogenous sense of time (*Delventhal et al., 2019*; *Schlichting et al., 2019*; *Yao and Shafer, 2014*), a large body of evidence supports the notion that s-LNvs are among the most critical neurons for the maintenance of endogenous circadian rhythms (reviewed by *Ahmad et al., 2021*). The 5th s-LNv was named because it was initially thought to be anatomically similar to the PDF positive s-LNvs in the larval brain (*Kaneko and Hall, 2000*; *Kaneko et al., 1997*; *Rieger et al., 2006*). However, subsequent work has suggested that the 5th s-LNv is likely functionally and anatomically more akin to the LNds (*Grima et al., 2004*; *Schubert et al., 2018*; *Yao and Shafer, 2014*).

The four PDF-expressing s-LNvs are characterized by a relatively simple morphology (*Figure 2A–C*; *Helfrich-Förster et al., 2007b*). Their cell bodies are located in the ventral brain, near the accessory medulla (AMe), into which they extend short neurites which likely receive input from photoreceptors (*Helfrich-Förster et al., 2002*; *Malpel et al., 2002*). These four sLNvs project dorsally to the posterior dorsal protocerebrum, where they turn toward the midline and form fine ramified termini that extend toward the midline (*Helfrich-Förster, 1997*; *Helfrich-Förster et al., 2007b*). These dorsal ramifications are thought to be the major site of s-LNv synaptic output but are also known to contain synaptic inputs (*Yasuyama and Meinertzhagen, 2010*). Work by *Schubert et al., 2018* showed that the 5th s-LNv (*Figure 2F*), which expresses the neuropeptide ion transport peptide (ITP) and choline acetyltransferase (*Johard et al., 2009*), an enzyme involved in the biosynthesis of acetylcholine, has more extensive ramifications within the dorsal brain and accessory medulla than the PDF expressing s-LNvs

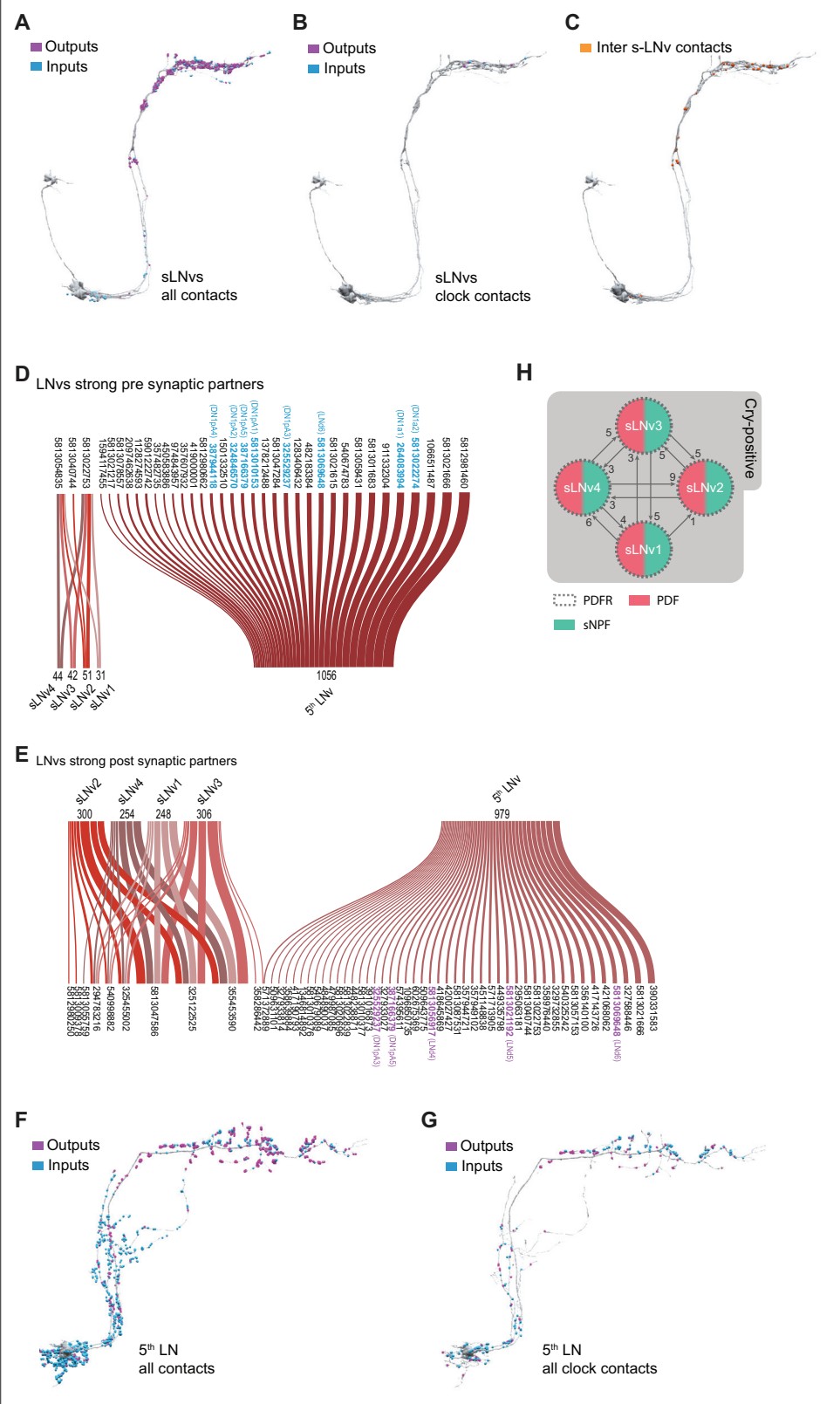

**Figure 2.** Connectivity patterns of the s-LNvs. (**A–C**). Synaptic connections of the four pdf + s LNvs, s-LNv_R_1 through s-LNv_R_4. Neuronal morphology is shown in gray. In A and B, inputs to the s-LNvs are shown in blue, outputs are shown in magenta. (**A**) All connections, including non-clock cells, (**B**) Connections to/from clock cells only, excluding connections to s-LNvs, (**C**) Connections among the four pdf + s LNvs. Input and output sides

*Figure 2 continued on next page*

*Figure 2 continued*

coincide and are indicated in orange. (**D–E**) Sankey diagram indicating the strong synaptic partners of all s-LNvs, including the 5th LNv. The total weight of synapses formed by each cell with its inputs (**D**) or outputs (**E**) is shown. (**D**) Presynaptic partners (inputs) of s-LNvs. No shared connections were found between the pdf + s LNvs and the 5th LNv. (**E**) Post synaptic partners (outputs) of s-LNvs. Only one cell (Body ID 571372889) receives synaptic input from the 5th LNv plus a pdf + s LNv. (**F–G**) Synaptic connections of the 5th LNv. (**F**) All connections, including non-clock cells, (**G**) connections to clock cells only. (**H**) Connectivity map of the four pdf + s LNvs indicating their peptide and receptor expression.

The online version of this article includes the following figure supplement(s) for figure 2:

**Figure supplement 1.** Connections of individual s-LNvs.

and provided evidence that they are anatomically more akin to LNds than to s-LNvs. Based on these findings, Schubert et al. suggested that the 5th sLNv be reclassified as a LNv (*Schubert et al., 2018*), a convention we follow for the remainder of this study. Based on this work, we predict that the 5th LNv within the hemibrain will be connectomically distinct from the PDF expressing s-LNvs.

The four PDF-expressing s-LNvs and the 5th LNv have been identified within the hemibrain dataset (*Figure 1B–C* and *Figure 2A and F*; *Scheffer et al., 2020*). Visualization of T-bars and postsynaptic densities within these neurons reveals that their ventral neurites, which innervate the accessory medulla, are biased toward receiving synaptic input (*Figure 2A and F*). Individual PDF-positive s-LNvs display relatively simple dorsal medial termini with relatively few branch-points that tend to run in parallel to neighboring branches (*Figure 2—figure supplement 1*). As previously described (*Yasuyama and Meinertzhagen, 2010*), these dorsal medial s-LNv termini contain both presynaptic and postsynaptic structures with the former significantly outnumbering the latter (*Figure 2A* and *Figure 2—figure supplement 1*). The dorsal termini of the 5th LNvs also contains both pre- and postsynaptic structures but appear to be less biased toward output compared to the PDF expressing s-LNvs (*Figure 2F*). Taken together, the four pdf +s LNvs form a total of 2238 synapses within the hemibrain volume: 505 inputs (postsynaptic densities) and 1733 outputs (projections onto postsynaptic densities). The single 5th s-LNv contains 1413 postsynaptic densities and forms synapses onto 1992 postsynaptic densities. Thus, the 5th LNv forms about four times the number of synapses of a single PDF-positive s-LNv (Figure 4C; *Supplementary file 1*, *Supplementary file 2*).

The four pdf +s LNvs display uniformity in their synaptic inputs for strong synaptic connections. Only three neurons in the hemibrain provide ten or more synapses onto at least one of these cells (*Figure 2D*), and only two of those three neurons form strong connections onto all four sLNvs (Figure 4D; *Supplementary file 1*). They also appear to be remarkably uniform in their patterns of strong synaptic output. Only 11 neurons receive 10 or more synapses from at least one *pdf* +s LNv (*Figure 2E*). Compared to the other LN classes, these s-LNvs are highly similar to each other in both their strong and medium-strength connections (Figure 4A, B) and form few but nearly uniform patterns of strong shared synaptic connections (Figure 4). Notably, within the hemibrain volume, the pdf +s LNvs are not strongly connected to other clock neurons, though they do form medium strength connections with one another (*Figures 1C and 2H*). The 5th LNv displays patterns of strong synaptic connectivity that are almost completely distinct from those of the four pdf +s LNvs, sharing a single strong output connection with only one of these cells (*Figure 2E*).

Though quite uniform in their patterns of strong synaptic input and outputs, the pdf +s LNvs do display some within-class differences in their patterns of weak and medium strength connections. Among the inputs targeting only one s-LNv, and mediated by only three synapses, is the HB-eyelet, a surprising finding given the long-held model that this external photoreceptor provides direct excitatory input onto s-LNvs and contributes to light entrainment of circadian rhythms (*Helfrich-Förster et al., 2002*; *Malpel et al., 2002*; *Rieger et al., 2003*), though recent work has indicated that the eyelet to s-LNv connection may be polysynaptic (*Li et al., 2018*). The sparse connectivity between the eyelet and the s-LNvs found within the hemibrain is consistent with recent work suggesting that the eyelet makes negligible contributions to light entrainment (*Alejevski et al., 2019*). Nevertheless, our analysis supports the notion that the pdf +s LNvs represent a uniform connectomic cell type with regard to patterns of strong synaptic connections, consistent with recent work revealing a uniformity in gene expression across the s-LNvs (*Ma et al., 2021*). A recent connectomic analysis of the larval brain reached similar conclusions regarding the uniformity of sensory inputs, anatomy, and local

connections among the four larval pdf +neurons (*Larderet et al., 2017*), which persist through meta-morphosis to become the adult s-LNvs (*Helfrich-Förster, 1997*).

## The LNds comprise several connectomic subclasses

As their name suggests, the somata of the dorsal lateral neurons (LNds) are situated dorsally relative to the LNvs, residing in the lateral cell body rind (*Helfrich-Förster et al., 2007b*; *Kaneko and Hall, 2000*; *Schubert et al., 2018*). All six LNds send dorsal medial projections across the superior proto-cerebrum and three send an additional projection toward the ventral brain (*Schubert et al., 2018*; *Figure 3* and *Figure 3—figure supplement 1*). Though consisting of only two more neurons than the PDF-positive s-LNv class, the LNds form seven times more synaptic connections (6149 postsynaptic densities and projections to 9590 PSDs; compare *Figures 2A–4C*) and are characterized by a much larger number of strong synaptic partners. There are 164 distinct neurons that receive strong synaptic inputs from at least one LNd (compared to 11 neurons for pdf +s LNvs; *Supplementary file 2*), and 86 distinct neurons provide strong synaptic input onto at least one LNd (compared to three for s-LNvs; *Supplementary file 2*).

In addition to forming more synapses with more neurons than the PDF-positive s-LNvs, the LNds are markedly less uniform in their patterns of strong synaptic connections (*Figures 3E, F and 4F, G*). There is not a single source of strong synaptic input or a single neural target of synaptic output that forms strong connections with all six LNds (*Figure 3E and F*, *Supplementary file 1*, *Supplementary file 2*). The maximum number of LNds that receive strong synaptic inputs from the same presynaptic neuron is three (*Figure 3*). The maximum number of LNds that form strong synaptic outputs onto the same neuronal targets is also three (*Figure 3F*). Based on these patterns of shared strong connectivity, there appear to be two connectomic groups within the LNds, each consisting of three neurons: LNds 1–3 and LNds 4–6. LNds within these two subgroups are more similar in their patterns of strong synaptic connectivity to each other than to LNds of the other group (*Figures 3E and F and 4F and G*). These two groups differ to a notable degree in the number of their synaptic connections, with the LNd 4–6 group forming approximately twice the number of synaptic connections (*Figure 4C*; *Figure 3—figure supplement 1* and *Supplementary file 1*, *Supplementary file 2*).

Examining the cellular morphology of the LNds within the hemibrain volume reveals that LNds 4–6 extend both dorsal-medial and ventral projections, whereas LNds 1–3 extend only dorsal-medial projections (*Figure 3—figure supplement 1*). As shown by *Schubert et al., 2018*, this indicates that LNds 4–6 are the Cryptochrome (Cry) expressing LNds, while LNds 1–3 correspond to the Cry-negative LNds (*Figure 3D*). Cry-positive and Cry-negative LNds differ in their expression of Pdf receptor (PdfR), with the Cry-positive LNds uniformly expressing PdfR and the Cry-negative LNds lacking receptor expression (*Im et al., 2011*), suggesting that these two groups of LNd are differentially sensitive to PDF released from the l-LNvs and s-LNvs.

Though the Cry-positive and Cry-negative subgroups of the LNds appear to be two distinct connectomic subtypes, patterns of strong synaptic connectivity suggest that the LNds can be further divided based on their patterns of strong synaptic connections. Within the Cry-positive LNds, LNd4, and LNd5 share 27 strong synaptic inputs with one another, but only seven strong connections with LNd6 (*Figure 3E*). Similarly, LNd4 and LNd5 share 33 strong synaptic targets with one another but only seven or fewer with LNd6 (*Figure 3F*). Thus, the Cry-positive LNds appear to consist of at least two connectomic subgroups, LNd4/LNd5 and LNd6. This connectomic distinction is consistent with differences in neuropeptide expression within the Cry-positive LNds, with two LNds expressing short neuropeptide F (sNPF) and the third expressing ion transport peptide (ITP), a feature it shares with the Cry expressing 5th LNv (*Johard et al., 2009*). A similar distinction can be made for the Cry-negative LNds, LNd2/LNd3 and LNd1, in terms of patterns of strong synaptic connections (*Figure 3E and F*). Based on these patterns of strong synaptic connections, the LNd class appears to consist of four distinct connectomic types. This number matches the number of transcriptomic classes of LNds recently revealed by single cell sequencing (*Ma et al., 2021*).

## The four connectomic groups of LNd clock neurons correspond to previously identified cellular and functional clock subsets

The LNds vary in their coupling patterns to the s-LNvs (*Yao and Shafer, 2014*) in addition to differences in their expression profiles (*Figure 3D*). The two sNPF positive LNds, which express Cry and

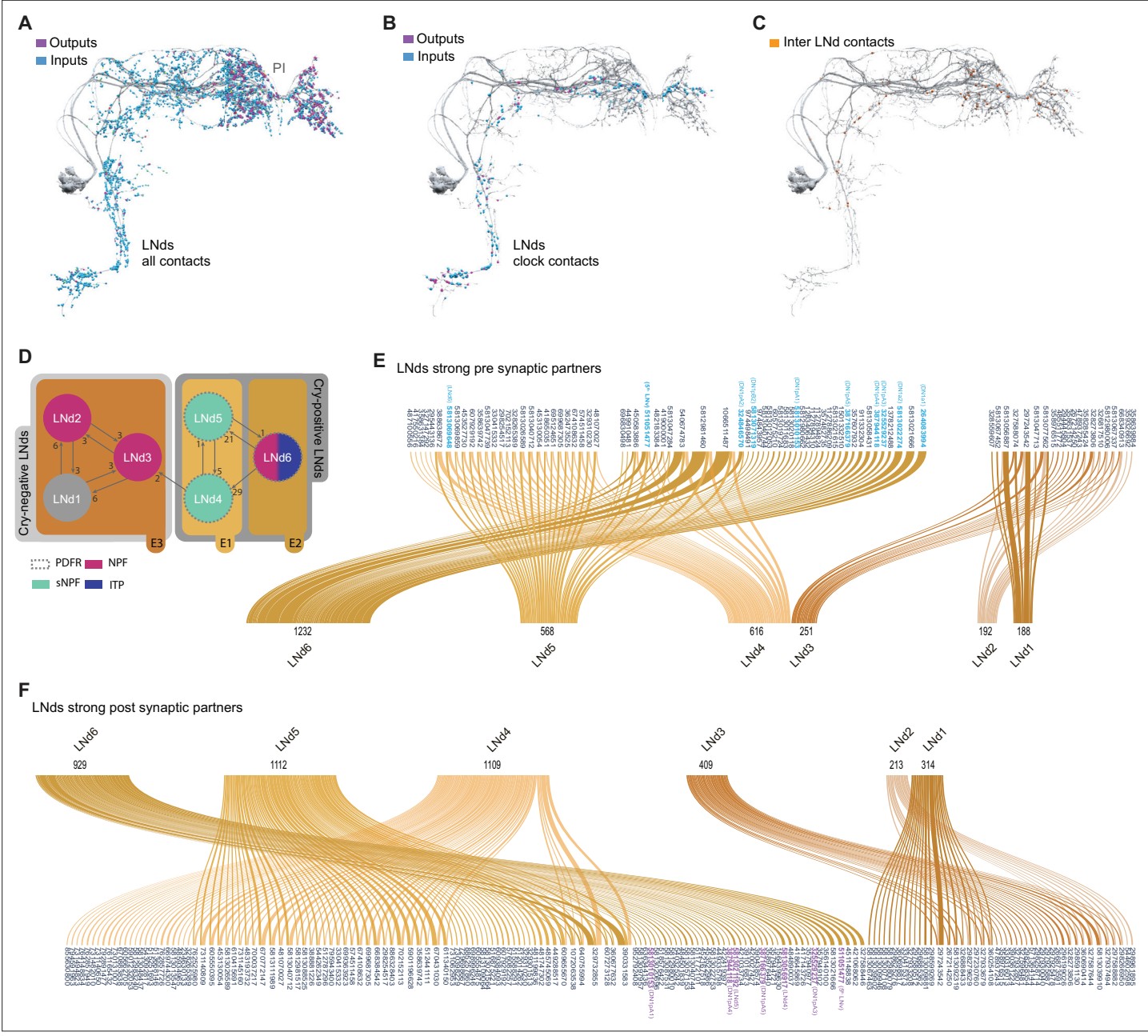

**Figure 3.** Connectivity patterns of the LNds. (**A–C**). Synaptic connections of the six LNds. Neuronal morphology is shown in gray. In A and B, inputs to the s-LNvs are shown in blue, outputs are shown in magenta. (**A**) All connections, including non-clock cells, (**B**) Connections to clock cells only, excluding connections to LNds. Input and output sides coincide and are indicated in orange. (**C**) Connections within LNds. (**D**) Connectivity map of the six LNds indicating their peptide and receptor expression. (**E–F**) Sankey diagram indicating the strong synaptic partners of all LNds. The total weight of synapses formed by each cell with its inputs (**E**) or outputs (**F**) is shown. (**E**) Presynaptic partners of LNds. No shared connections were found between LNd1-3 and LNd4-6. Most of the shared connections are between LNd4 and LNd5 (X cells provide strong inputs to both). (**F**) Postsynaptic partners of the LNds.

The online version of this article includes the following figure supplement(s) for figure 3:

**Figure supplement 1.** Connections of individual LNds.

**Figure supplement 2.** Connections of E cell classes.

PdfR, remain tightly coupled to the s-LNvs when the circadian clock in the latter neurons is slowed down (*Yao and Shafer, 2014*). The sNPF expressing LNds were termed E1 oscillators based on this tight temporal coupling to the s-LNvs (*Yao and Shafer, 2014*), which also express sNPF (*Johard et al., 2009*). The NPF/ITP expressing LNd, despite being receptive to PDF released from the LNvs, does not

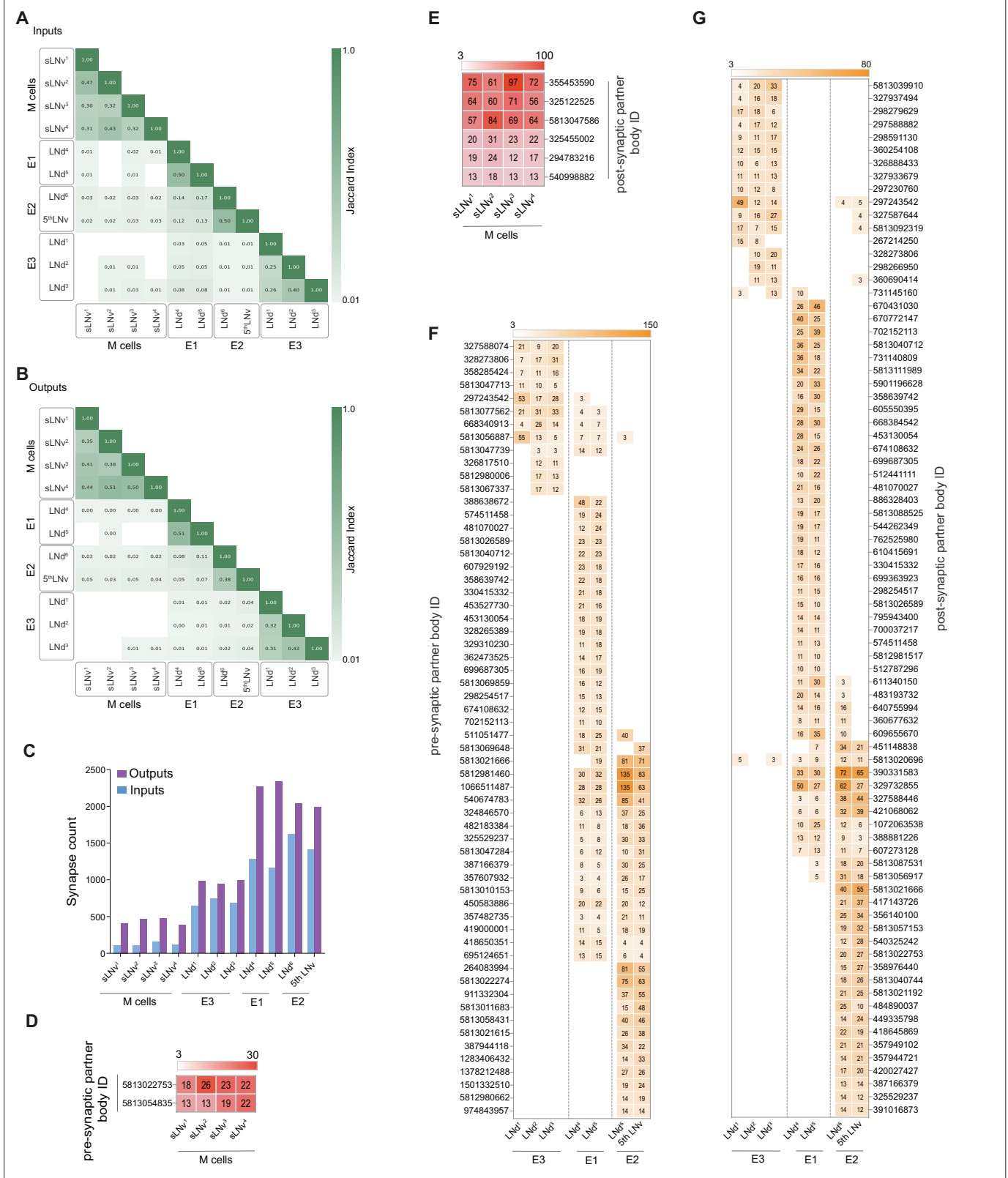

**Figure 4.** M cells are homogeneous while E cells can be clustered in three distinct groups. (**A–B**) Jaccard indices for overlap in synaptic partners of M and E cells. Only includes synaptic partners that make medium or strong connections. Higher index values indicate more similarity in either inputs (**A**) or outputs (**B**). (**C**) Total input and output Synapse counts for M and E cells. (**D–E**) Strong shared connections of the four pdf +s LNvs. Only cells that share one connection with at least two M cells are shown. The strength (weight) of the connection is indicated. Only medium and strong connections

*Figure 4 continued on next page*

*Figure 4 continued*

are included. (**D**) The two cells that send strong connections to at least two M cells send strong connections to all four. (**E**) The six cells that receive strong connections from at least two M cells receive strong connections from all four. (**F–G**) Strong shared connections of the six LNds plus the 5th LNv (collectively referred to as E cells). Only cells that share a strong connection with at least two E cells are shown. The strength (weight) of the connection is indicated. Only medium and strong connections are included. (**F**) Cells that send strong connections to at least two E cells are included in the heatmap. (**G**) Cells that receive strong connections from at least two E cells are included in the heatmap.

The online version of this article includes the following figure supplement(s) for figure 4:

**Figure supplement 1.** Comparison of medium and strong inputs of 5th s-LNv relative to the pdf +s LNvs and the LNds.

**Figure supplement 2.** Comparison of medium and strong outputs of 5th s-LNv relative to the pdf +s LNvs and the LNds.

display coupling to the slowed s-LNv clocks and was grouped with the ITP-expressing 5th s-LNv as E2 oscillators (*Yao and Shafer, 2014*). The remaining LNds, which consist of the Cry-negative LNds, were classified as E3 oscillators, which are neither PDF receptive nor coupled to s-LNvs with modified clock speeds (*Yao and Shafer, 2014*).

The Cry-positive LNds, therefore, appear to consist of two functional types; two LNds making up E1 and a single LNd corresponding to E2. *Schubert et al., 2018* established the anatomical hallmarks of these two Cry-positive LNds, with the single E2 LNd sending more extensive ventral projections compared to the E1 LNds. Examination of hemibrain LNd volumes indicates that LNd6 corresponds to LNd-E2 (compare *Figure 3—figure supplement 1*), and that LNd4 and LNd5 are the E1 LNds. This LNd5/4 (E1) and LNd6 (E2) distinction mirrors the patterns of shared strong synaptic connectivity displayed by the Cry-positive LNds (*Figure 3E and F*). Therefore, functionally distinct LNd subgroups share patterns of strong synaptic inputs and outputs.

The 5th LNv, which shares cellular and anatomical similarities with LNd6 (*Johard et al., 2009*; *Schubert et al., 2018*), was assigned as a member of the E2 functional class by *Yao and Shafer, 2014*, based on the absence of coupling to timekeeping within the PDF positive s-LNvs, despite receptivity to PDF and their role in the generation of evening activity (*Grima et al., 2004*). We, therefore, asked if LNd6 and the 5th LNv display similar patterns of synaptic connectivity. Indeed, these two neurons displayed significant overlap in their strong synaptic connectivity, both for synaptic inputs and targets (*Figure 4F and G*). Thus, LNd6 and the 5th LNv are more alike in their patterns of strong synaptic connectivity than they are to other lateral neuron subtypes in addition to sharing neuropeptide expression and coupling mode with the PDF expressing s-LNvs.

The patterns of strong clock neuron connectivity within the hemibrain led us to hypothesize that the recognized functional subtypes, M (PDF-positive s-LNvs), E1 (LNd4 and LNd5), E2 (LNd6 and 5th LNv), and E3 (LNd1, LNd2, and LNd3) display distinct patterns of synaptic connectivity. To test this hypothesis, we expanded our analysis to include both strong and medium strength synaptic connections, which together account for the vast majority of connections made by the identified clock neurons within the hemibrain dataset. To examine affinities between individual identified clock neurons in our expanded consideration of connectivity, we used the Jaccard index (*Figure 4A and B*), a coefficient of similarity between two sets that ranges between zero and one, the former value indicating no overlapping synaptic partners and latter indicating complete overlap (see Methods). Jaccard indices for both synaptic inputs (*Figure 4A*) and outputs (*Figure 4B*) provide strong evidence that the LN classes can be divided into four connectomic groups: The four PDF positive s-LNvs (M group; Figure 6), two of the Cry-positive LNds (LNd4 and LNd5; E1 group; Figure 7), LNd6 and the 5th LNv (E2 group; Figure 8 and *Figure 3—figure supplement 2*), and the Cry- negative LNds (LNd1, LNd2, and LNd3; E3 Group; Figure 9), conforming to the functional divisions hypothesized by *Yao and Shafer, 2014*.

These functional LN divisions also have different numbers of synaptic connections and different ratios of input to output synapses. The M group displays the smallest number of synapses, E3 an intermediate number, and E1 and E2 form roughly twice the number of connections displayed by E3 (*Figure 4C*). The M and E1 groups also appear distinct from the E2 and E3 groups in the balance of output to input synapses. Though all the identified LNs form more synaptic outputs than inputs, output synapses in the M and E1 groups make up a larger proportion of total synaptic connections when compared to the E2 and E3 groups (*Figure 4*). Thus, the connectomic subgroups that emerged from our analysis of strong synaptic connectivity (*Figures 2–4*) are further supported by the combined analysis of strong and medium strength connectivity (*Figure 4—figure supplements 1 and 2*) and conform to previously hypothesized functional and cellular subgroups.

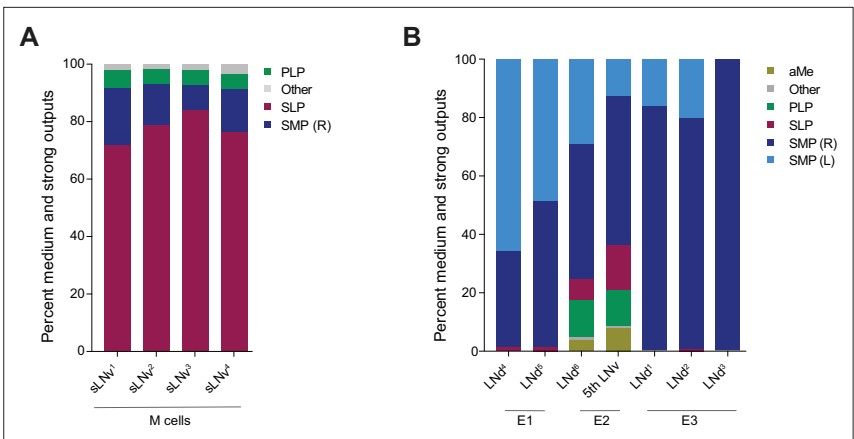

**Figure 5.** Neuropils innervated by M and E cells. (**A–B**). Percentage of connections located in each of the indicated neuropils. Medium and strong connections are included. (**A**) Neuropils in which the outputs of each of the four pdf + s LNvs are located. (**B**) Neuropils in which the outputs of each of the LNds and the 5th s-LNv are located.

## The functional/connectomic subgroups of LN clock cells display distinct synaptic output pathways

In animals, the brain's endogenous central circadian clock drives myriad behavioral, physiological, and endocrine rhythms. Synaptic connections between the central clock network and neurons outside the timekeeping network are thought to mediate daily signals from the circadian system to the neural centers responsible for driving daily physiological and behavioral changes. Though specific output pathways linking the clock to specific endocrine and sleep control centers have been previously described in the fly (reviewed by *King and Sehgal, 2020*; *Shafer and Keene, 2021*), there is still a great deal to be learned about circadian output signaling. For example, do the various functional groups of clock neurons converge on the same synaptic targets to shape the timing of the same daily outputs, as was recently described for arousal promoting pathways linking the LNs to dopaminergic neurons modulating the ellipsoid body (*Liang et al., 2019*)? Alternatively, do functional clock cell subgroups generally synapse onto distinct neural targets to produce uniquely phased outputs of different behavioral, physiological, and endocrine rhythms, as recently described for sleep modulating dorsal clock neurons (*Guo et al., 2018*; *Lamaze et al., 2018*)? We examined the patterns of synaptic outputs of the LN oscillator subgroups to determine how they coordinate the myriad circadian outputs.

As previously described (*Schubert et al., 2018*) M, E1, E2, and E3 subgroups can be distinguished by their patterns of neuropil innervation. As expected, the M, E1, E2, and E3 subgroups differ significantly in terms of the brain areas where the majority of their neurites form chemical synapses. Though the M and E subgroups form output synapses within both the superior lateral protocerebrum (SLP) and superior medial protocerebrum (SMP), the M subgroups form the majority of their output synapses within the SLP, whereas the E groups form the majority of their output synapses within the SMP (*Figure 5A and B*). Though all seven E cells terminate largely within these two neuropils, they fall, once again, into three classes with regard to their neuropil innervation. For example, E1 forms more output synapses on cells of the anterior ventrolateral protocerebrum (AVLP) than E2 or E3, E2 forms more synapses within the accessory medulla than do E1 or E3, and E3 forms little to no synapses on neurons associated with either of these neuropils (*Figure 5B*).

The M subgroup's shared strong synaptic outputs represent only three distinct cell types (*Figure 6*). Among these are five neurons within the SLP, which represent only two cell types: three neurons annotated as SLP-316-R (*Figure 6A*) and two neurons annotated as SLP-403-R (*Figure 6B*). The sixth strong shared output is an SMP-associated neuron named SMP232-R (*Figure 6C*). The SLP-316-R neurons strongly resemble DN1ps currently missing from the hemibrain annotation, which sends ventral projections alongside the dorsal projections of M cells and were named ventral-contralateral DN1ps by *Lamaze et al., 2018* (*Figure 6A* and *Figure 6—figure supplement 1*). Jaccard indices support the conclusion that these strong shared targets of M output are three distinct cell types

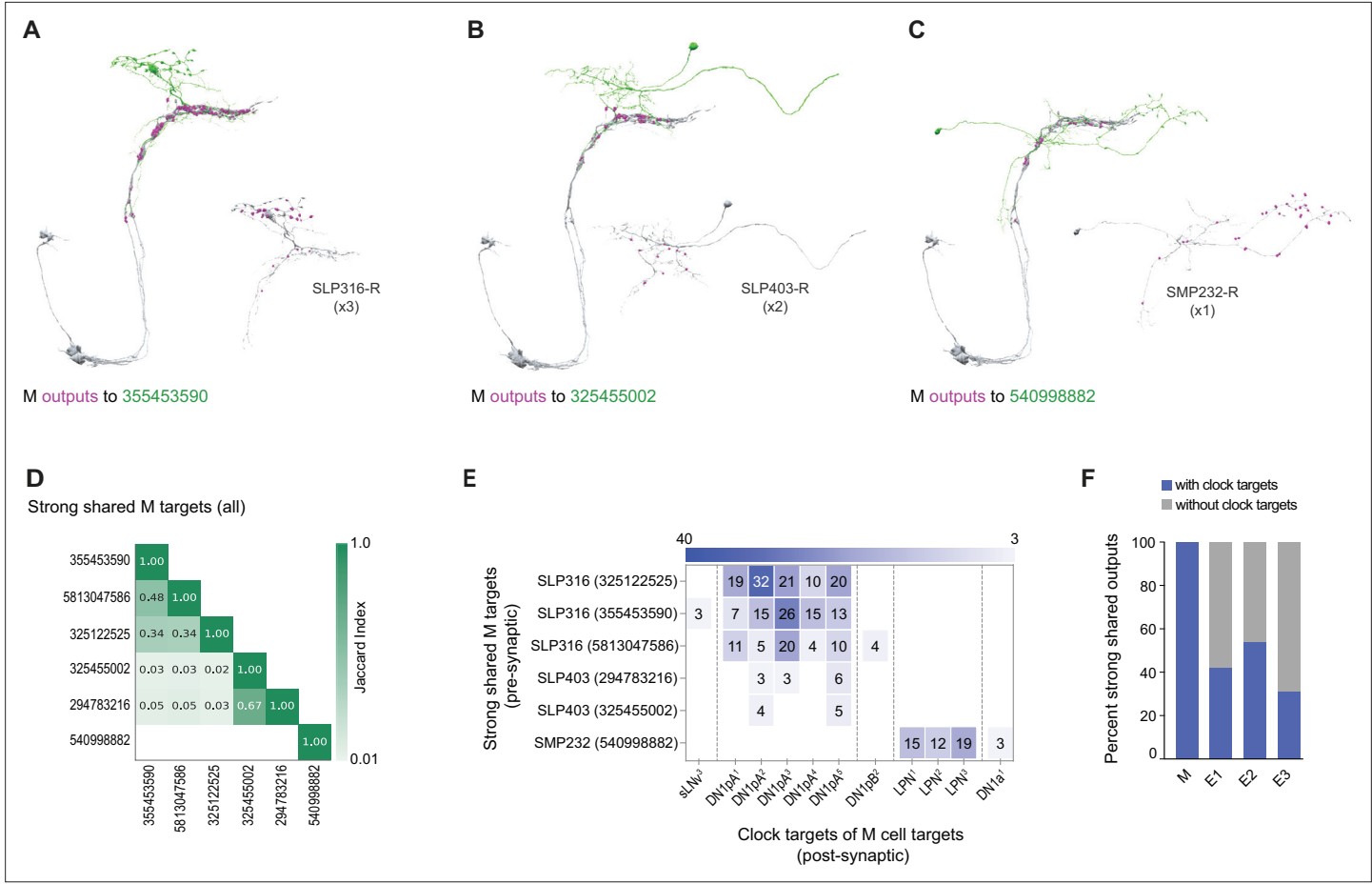

**Figure 6.** Strong shared outputs of M cells. (**A–C**) The six cells that are strong shared outputs of the four M cells involve three different neuronal types (SPL316-R, SLP403-R, and SMP232-R). The M cells outputs onto each representative neuron of each type are shown in magenta. Representative target neurons are shown in green. (**A**) M cells contact three SPL316 neurons. Left, all four M cells are shown in gray and their contacts to 355453590 (neuronal morphology shown in green) are shown in magenta. Right: 355453590 is shown in gray and its outputs to clock cells are shown in magenta. (**B**) M cells contact two SPL403-R neurons. Left, all four M cells are shown in gray and their contacts to 325455002 (neuronal morphology shown in green) are shown in magenta. Right: 325455002 is shown in gray and its outputs to clock cells are shown in magenta. (**C**) M cells contact one SMP232-R neuron. Left: all four M cells are shown in gray and their contacts to 325455002 (neuronal morphology shown in green) are shown in magenta. Right: 325455002 is shown in gray and its outputs to clock cells are shown in magenta. (**D**) Jaccard index of the six M cell shared output cells. The index is based on the similarity of their outputs, the more similar their outputs are the higher the index value. Y and x-axis indicate the cell body ID of each of the six cells. Their neuronal type is indicated to the left of the body ID on the y-axis. Only indices ≥ 0.01 are shown. (**E**) All strong shared outputs of M cells in turn contact clock neurons. On the x-axis, the clock neurons that receive contacts from each cell are indicated. The values on the cells represent the weight of each connection. Medium and strong connections are included. (**F**) Percent of strong shared outputs of each neuronal class that in turn sends contacts to clock cells. Medium and strong connections are included. E1=LNd4 and LNd5, E2=LNd6 and the 5th s-LNv, E3=LNd1, LNd2, and LNd3.

The online version of this article includes the following figure supplement(s) for figure 6:

**Figure supplement 1.** Connections of individual strong shared outputs of s-LNvs.

**Figure supplement 2.** Neuroanatomy of the three SLP316 cells and identified DN1s.

(*Figure 6D*). Remarkably, all of the shared strong output targets of the M group form synapses back onto identified clock neurons (*Figure 6A–C*), including strong synaptic connections onto DN1ps and LPNs (*Figure 6E*). Thus, all the strong and shared output targets of the M group project back to the clock neuron network, thereby implicating them as nodes within the clock neuron network, either as currently unidentified clock-containing neurons or non-clock-containing neurons. The M group appears to be unique in the extent to which its strong shared synaptic targets reenter the clock neuron network, with only half or fewer of strong E1, E2, or E3 targets recurring to the clock network in this fashion (*Figure 6F*).

The E1 subgroup is characterized by many more strong/shared synaptic targets than the M subgroup (compare *Figure 4E and G*) and these targets appear, based on Jaccard indices, to include many distinct cell types (*Figure 7D*). A minority of these E1 targets (16 out of 38) form output synapses onto identified clock neurons within the hemibrain volume. Thus, the majority of the strong shared synaptic outputs of E1 do not immediately project back to the clock neuron network, in contrast to the uniform recurrence of strong shared M targets to the clock network (*Figure 6A–E*). Several of the targets receiving the strongest E1 synaptic input do form strong synaptic connections back onto clock neurons, particularly onto the E1 LNds themselves, in addition to LPNs and DN1as (*Figure 7E*). Notably, the strong shared targets that recur to the clock neuron network do not display the anatomical hallmarks of any of the currently unannotated clock neurons in the hemibrain dataset (e.g. *Figure 7A–C*). It, therefore, appears that neurons that do not themselves possess molecular clocks might nevertheless reside within the central timekeeping network and mediate polysynaptic connections between clock neurons.

The E2 subgroup, like E1, forms strong shared synapses on a much larger number of neuronal targets than the M group (compare *Figure 4E and G*) and these targets are a diverse array of cell types. Approximately half (14 out of 26) of the strong and shared E2 targets form synaptic connections onto identified clock neurons within the hemibrain (*Figure 6F*, *Figure 8A–C and E*). Compared to the recurrent E1 clock connections, E2 forms synapses on identified clock neurons more broadly, forming connections with all currently identified clock neuron classes in the hemibrain volume, with the exception of the DN1a class (*Figure 8E*). Thus, the E2 group appears not only to represent a hub of strong direct inter-connectivity between identified clock neuron classes (*Figure 1C*), but also provides additional synaptic clock network inputs via their strong and shared synaptic targets (*Figure 8E*). As for E1, strong shared targets of E2 recurring to the clock network include neurons that do not share an obvious anatomical affinity for currently unidentified clock neurons within the hemibrain dataset (e.g. *Figure 8A–C*). This suggests, once again, that the circadian timekeeping network of *Drosophila* includes neurons that do not themselves have a molecular clock.

Compared to the E1 and E2 groups, E3 forms approximately half the number of synaptic connections (*Figure 4C and G*). With regard to its shared strong outputs, the E3 subgroup forms strong shared synaptic connections onto only 16 neurons (*Figures 3F and 4G*), five of which form synapses onto identified clock neurons within the hemibrain (*Figure 9E*). The synapses that those five neurons form back onto clock neurons are limited to LNds and LPNs, with most connections forming back onto E3 LNds themselves (*Figure 9A–C and E*). This makes E3 similar to the E1 group in that they are characterized by strong shared output targets that form synapses directly back onto their E3 inputs (*Figure 7E* and *Figure 9E*). Once again, the strong and shared output targets recurring to the clock network do not bear the anatomical hallmarks of clock neurons missing from the annotation (*Figure 9A–C*).

Taken together, our analysis indicates that the strong synaptic outputs of the four functional/connectomic LN subgroups, M, E1, E2, and E3, diverge onto distinct neural targets (*Figure 4*). Furthermore, many of these synaptic targets synapse back onto neurons within the clock neuron network, including all of the shared and strong output targets of the M subgroup. In contrast, approximately half of the synaptic targets of synaptic output from the E subgroups do not immediately recur to the clock network (*Figure 6F*). Not only does this implicate specific neuronal targets as conduits of circadian output, but it also suggests that neurons not previously identified as 'clock neurons' represent integral nodes within the neural network from which endogenous circadian timekeeping emerges (see Discussion).

## Discussion
### The clock neuron network within the hemibrain volume

Before discussing our connectomic analysis of the LN clock neuron classes, it is important to acknowledge the limitations of the dataset. The hemibrain represents a single hemisphere of the central brain and therefore does not contain all connections from the clock neuron cell bodies it contains. Several classes of clock neurons, including the large LNvs, the 5th LNv, the LNds, and DN1ps, project across the midline and form connections in the contralateral hemisphere (*Helfrich-Förster et al., 2007b*). Many of these connections are expected to reside outside of the hemibrain volume. Our estimates of

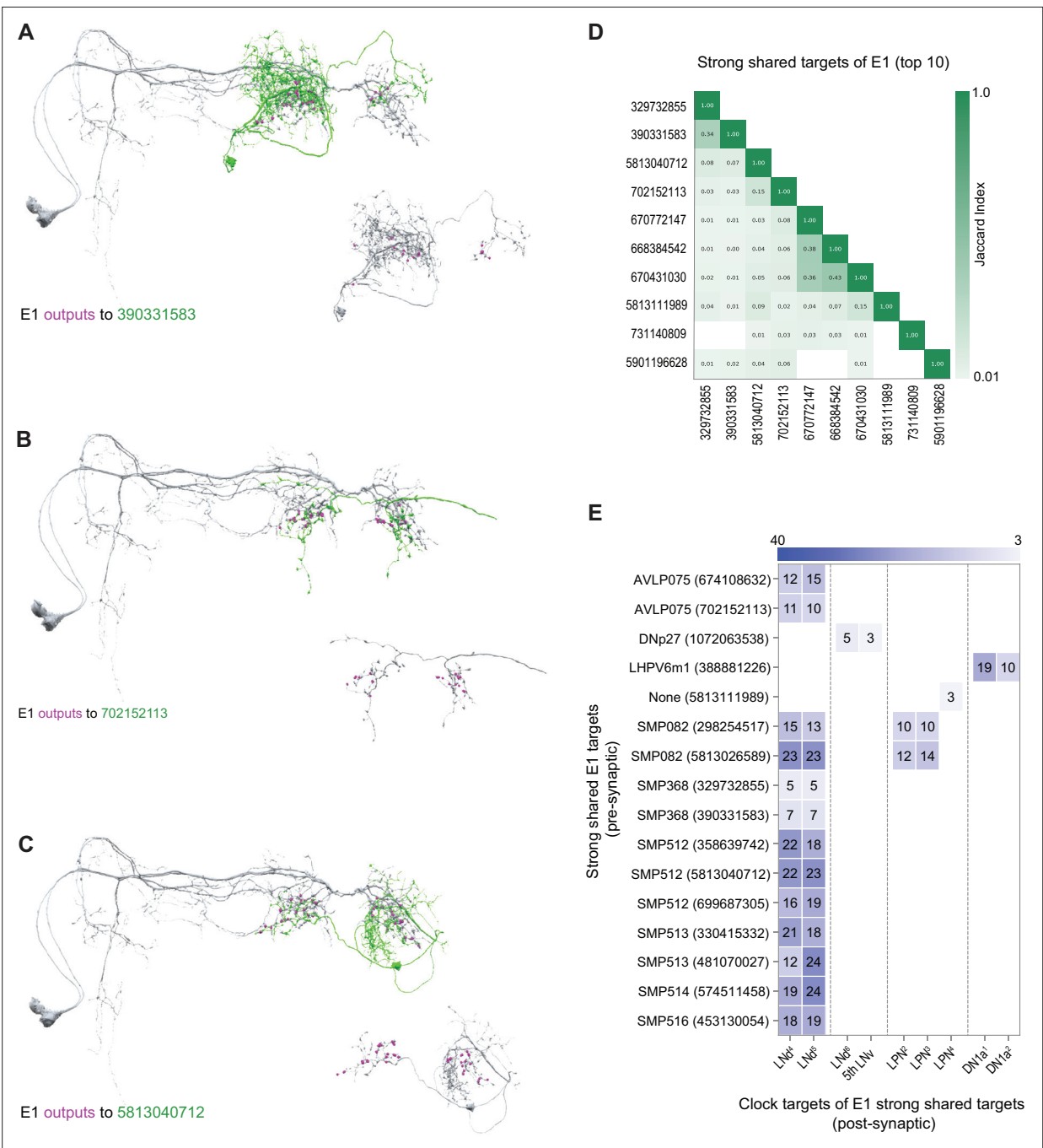

**Figure 7.** Strong shared outputs of E1. (**A–C**) The cells that are strong shared outputs of E1 involve multiple neuronal types. E1 outputs onto three representative neurons are shown. E1 neurons (LNd4 and LNd5) are shown in gray in the left panels, where representative output neurons are shown in green. On the right panels, each representative cell is shown in gray and its contacts to clock cells are shown in magenta (**A**) E1 cells contact two SMP368-R neurons. Left, E1 neurons (shown in gray) contacts to SMP368-R neuron 390331583 (shown in green) are shown in magenta. (**B**) E1 cells contact two AVLP075-R neurons. Left, E1 neurons (shown in gray) contacts to AVLP075-R neuron 702152113 (shown in green) are shown in magenta. (**C**) E1 cells contact two SMP315-R neurons. Left, E1 neurons (shown in gray) contacts to SMP315-R neuron 5813040712 (shown in green) are shown in magenta. (**D**) Jaccard indices indicating overlap among the output synaptic partners of the top 10 E1 strong shared output cells. The index is based on the similarity of their outputs, the more similar their outputs are the higher the index value. Y and x-axis indicate the cell body ID of each of the six cells. Only indices ≥ 0.01 are shown. (**E**) Most shared outputs of E1 cells that contact clock cells send strong contacts to both E1 neurons. On the x-axis, the clock neurons that receive contacts from each cell are indicated. The values on the cells represent the weight of each connection. Medium and strong connections are included.

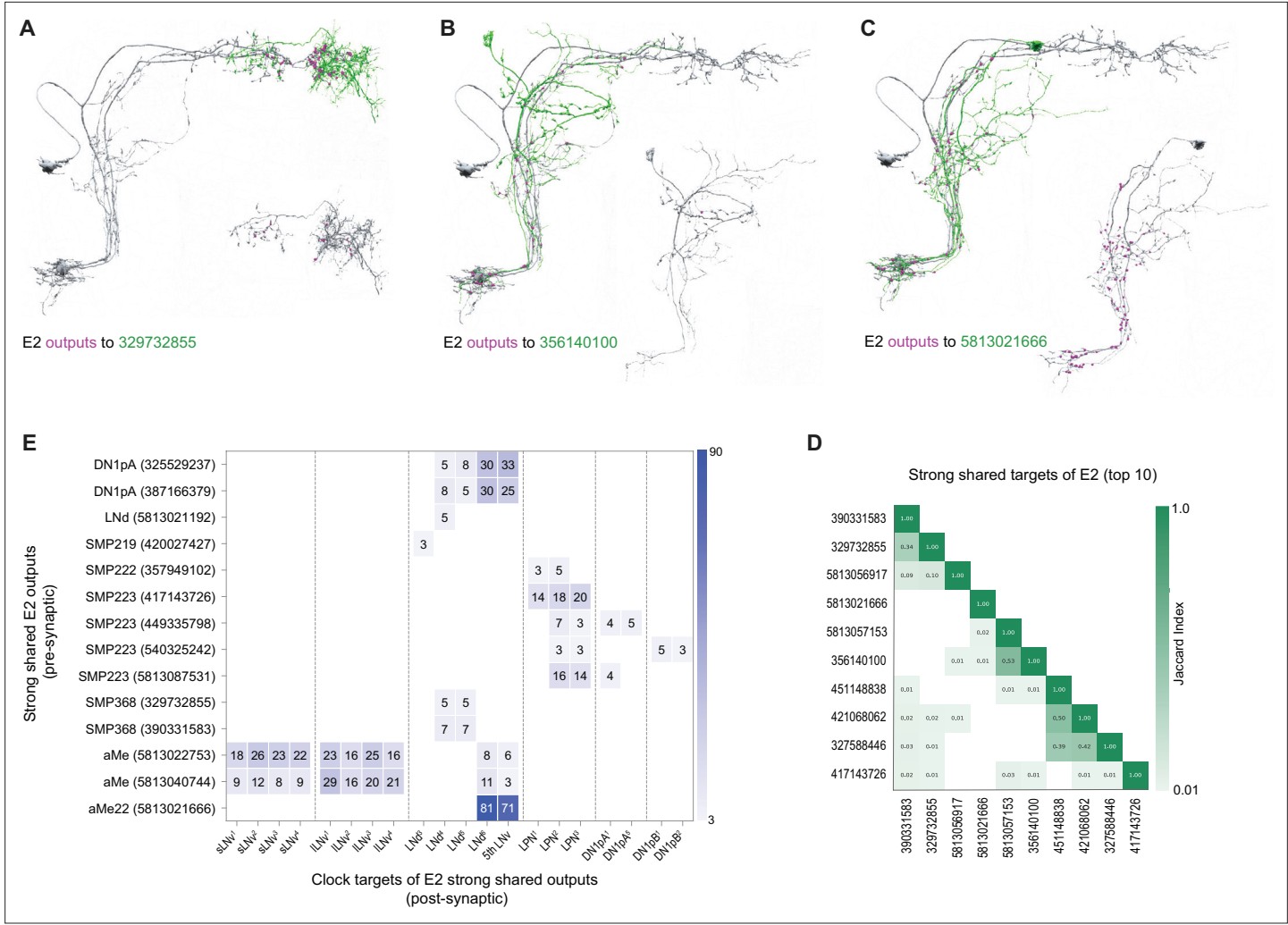

**Figure 8.** Strong shared outputs of E2. (**A–C**) The cells that are strong shared outputs of E2 involve multiple neuronal types. E2 outputs onto three representative neurons are shown. E2 neurons (LNd6 and the 5th s-LNv) are shown in gray in the left panels, where representative output neurons are shown in green. On the right panels, each representative cell is shown in gray and its contacts to clock cells, if present, are shown in magenta (**A**) Left, E2 neurons (shown in gray) contacts to SMP368 neuron 329732855 (shown in green) are shown in magenta. (**B**) Left, E2 neurons (shown in gray) contacts to SLP249 neuron 356140100 (shown in green) are shown in magenta. (**C**) Left, E2 neurons (shown in gray) contacts to aMe22 neuron 5813021192 (shown in green) are shown in magenta. (**D**) Jaccard indices for the outputs of the top 10 E2 strong shared output cells. The index is based on the similarity of their outputs, the more similar their outputs are the higher the index value. Y and x-axis indicate the cell body ID of each of the ten cells. One of the strong shared outputs of E2 is LNd4. Only indices ≥ 0.01 are shown. (**E**) Shared outputs of E2 that contact clock cells contact different clock subclasses. On the x-axis, the clock neurons that receive contacts from each cell are indicated. The values on the cells represent the weight of each connection. Medium and strong connections are included.

inter-clock connectivity will therefore not include these contralateral connections. Furthermore, recent work suggests that electrical synapses within the clock neuron network likely contribute to circadian timekeeping (*Ramakrishnan and Sheeba, 2021*), but these are not visible in the hemibrain dataset (*Scheffer et al., 2020*). In addition, neuropeptides can be released non-synaptically (e.g. *Yasuyama and Meinertzhagen, 2010*), act over long distances, and such paracrine signaling likely contributes to circadian timekeeping in the brain (e.g. *Maywood et al., 2011*). The connectomic analysis presented here does not include such peptidergic connectivity. Thus, our analysis underestimates connectivity. In addition, clock neuron classes undergo daily morphological changes that are most likely accompanied by changes in the number and locations of synaptic connections (*Duhart et al., 2020*; *Fernández et al., 2008*; *Petsakou et al., 2015*; *Song et al., 2021*). Thus, the hemibrain represents only one time-point, in the early morning, within a cycle of changing synaptic connections (*Scheffer et al., 2020*). Finally, the hemibrain dataset represents a single female fly, whereas the vast majority of experiments

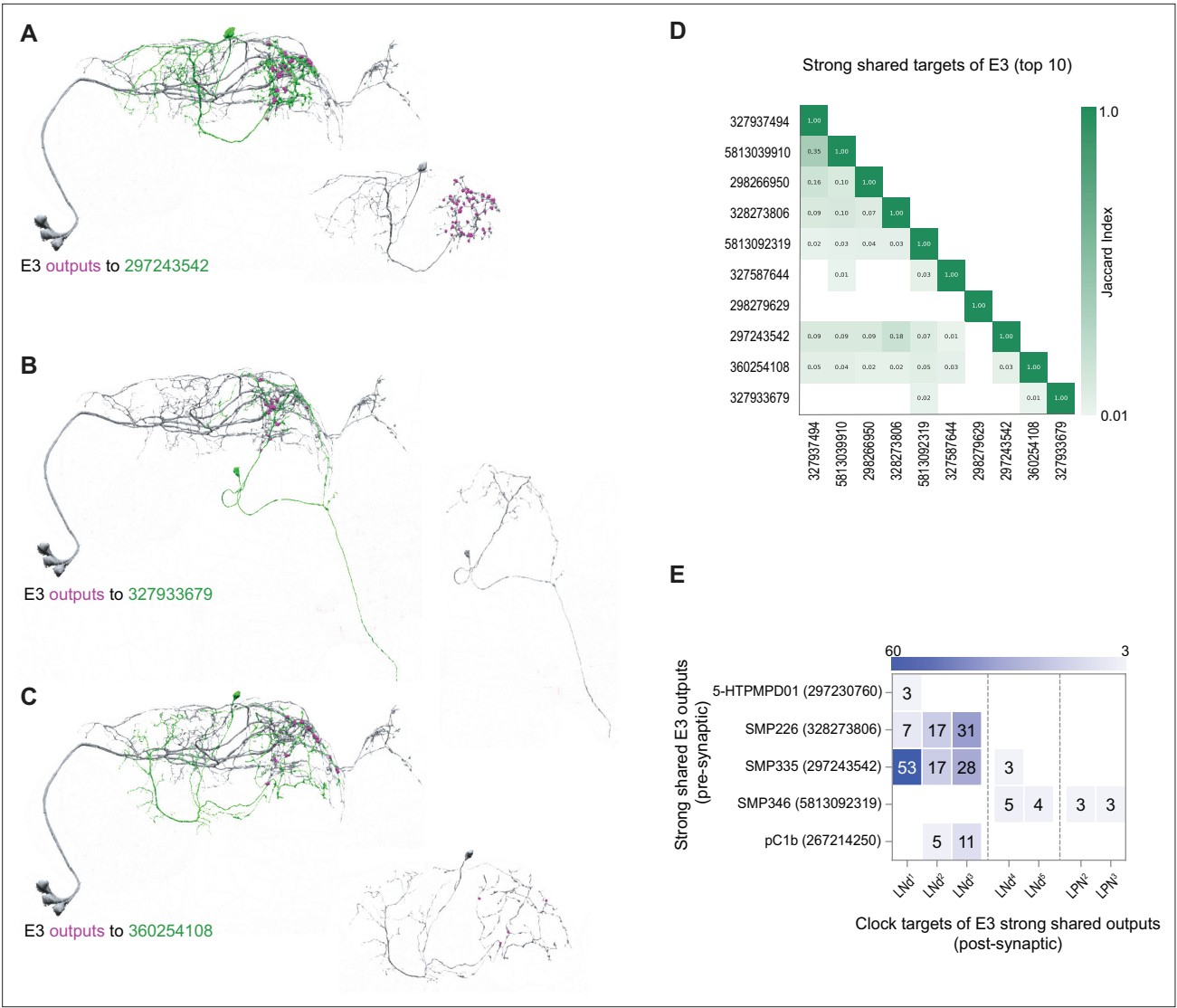

**Figure 9.** Strong shared outputs of E3. (**A–C**) The cells that are strong shared outputs of E3 involve multiple neuronal types. E3 outputs onto three representative neurons are shown. E3 neurons (LNd1, LNd2, and LNd3) are shown in gray in the left panels, where representative output neurons are shown in green. On the right panels, each representative cell is shown in gray and its contacts to clock cells are shown in magenta. (**A**) Left, E3 neurons (shown in gray) contacts to SMP335 neuron 297243542 (shown in green) are shown in magenta. (**B**) Left, E3 neurons (shown in gray) contacts to SMP486 neuron 327933679 (shown in green) are shown in magenta. (**C**) Left, E3 neurons (shown in gray) contacts to SMP334 neuron 360254108 (shown in green) are shown in magenta. (**D**) Jaccard indices for the outputs of the top 11 E3 strong shared output cells. The index is based on the similarity of their outputs, the more similar their outputs are the higher the index value. Y and x-axis indicate the cell body ID of each of the cells. Only three cells are strong shared outputs of all E3 cells (E3a+E3 b). 8 cells are strong shared targets of E3a only. Only indices ≥ 0.01 are shown. (**E**) Shared outputs of any two E3 cells that contact clock cells. On the x-axis, the clock neurons that receive contacts from each cell are indicated. The values on the cells represent the weight of each connection. Medium and strong connections are included.

on the neural basis of circadian timekeeping have employed male files. Thus, unexpected connectivity may be due to differences between the well-characterized clock neuron network of males and the less well-studied female network.

Though, we are focusing our analysis on the LN clock neuron classes, we have omitted the l-LNvs. We have done so because the synaptic outputs of the l-LNvs are thought to occur in the contralateral accessory medulla (AMe) and much of their synaptic inputs are thought to be located in regions of the medulla that are not included in the hemibrain volume (Helfrich-Forster, Yoshii, *Helfrich-Förster et al., 2007a*). We note, in this context, that the l-LNvs do not appear to make significant contributions to endogenous circadian timekeeping (*Shafer et al., 2002*; *Shafer and Taghert, 2009*; *Yang*

*and Sehgal, 2001*), which is the phenomenon we seek to illuminate here. Despite these limitations, the organizational principles uncovered in this study provide insights into clock network organization and are sure to generate a significant number of testable hypotheses regarding network function (see below).

## Connectomic classes of LN clock neuron mirror previously proposed functional and molecular LN sub-groups

The lateral neuron classes are critical to maintain endogenous circadian rhythms (*Helfrich-Förster, 1998*; *Renn et al., 1999*). An examination of genetic mosaics suggested that, within the LNs, the PDF expressing LNvs drive the morning peak of daily activity, whereas PDF negative LNds and 5th LNv drive the evening peak, leading to the designation of the former neurons as 'Morning (M) Cells' and the latter as 'Evening (E) Cells' (*Grima et al., 2004*; *Stoleru et al., 2004*). Interestingly, light appears to modulate the coordination between the morning and evening oscillators (*Chatterjee et al., 2018*; *Lamba et al., 2018*). Though named based on its anatomical similarity to the other LNv clock neurons in the larval brain (*Kaneko and Hall, 2000*; *Rieger et al., 2006*; *Schubert et al., 2018*), there are now numerous observations suggesting that the 5th-LNv is functionally and anatomically distinct from the PDF expressing s-LNvs in the adult brain: the four pdf +sLNvs are functionally associated with morning activity whereas the 5th LNv is associated with evening activity (*Grima et al., 2004*). Furthermore, the 5th LNv shares connectivity patterns, neuropeptide expression, and features of cellular anatomy with one of the Cry-expressing LNds (*Johard et al., 2009*; *Liang et al., 2017*; *Schubert et al., 2018*; *Yao and Shafer, 2014*). Our connectomic analysis further supports the conclusion that the 5th LNv is an LNd-like clock neuron that bears little resemblance, functionally or anatomically, to the PDF expressing s-LNvs.

Functional and anatomical analysis suggests that the LN neurons can be divided into four functional classes, M, E1, E2, and E3 and that the patterns of strong connectivity displayed by the LNs are in striking concordance with these divisions (*Yao and Shafer, 2014*). Our connectomic analysis indicates that the functional differences characterizing these four groups of neurons appear to be written in the connectome: each receives a unique combination of strong synaptic inputs and, in turn, forms strong synaptic connections onto distinct postsynaptic targets. These results suggest that the functional subsets of LNs likely drive distinct behavioral and physiological outputs, rather than converging onto the same premotor, sleep, or endocrine centers.

In addition, our analysis supports the existence of an additional subgroup within the LNs, suggesting that the Cry-negative E3 class likely consists of two subgroups: LNd2 and LNd3, which share a large proportion of their strong synaptic inputs and outputs, and LNd1, which shares significantly fewer strong connections with the other two Cry-negative LNds (*Figure 3E and F*). Placing LNd1 into its own LNd subgroup aligns our connectomic LN divisions with a recent transcriptomic analysis of the clock neuron network that divided the LNds into four clusters: The two sNPF and Cry expressing LNds (LNd4 and LNd5, i.e. E1), the single ITP and Cry expressing LNd (LNd6, i.e. the E2 LNd), two NPF expressing LNds lacking Cry expression, and a single Cry-negative LNd that lacks NPF (*Ma et al., 2021*).

## The M group forms very few synaptic connections with other identified clock neuron classes in the hemibrain dataset

The s-LNvs of the M group, though they form medium strength connections with one another, are almost completely isolated from the other identified clock neurons within the hemibrain dataset (*Figures 1C and 2H*). Though this picture may ultimately underestimate of inter-clock connectivity if the strong shared SLP targets of M output (*Figure 6A*) are determined to be the DN1ps currently missing from the annotation (*Figure 6—figure supplement 2*). The predicted synaptic inputs to the M group from the DN1ps (*Collins et al., 2012*; *Fernandez et al., 2020*; *Guo et al., 2016*; *Yadlapalli et al., 2018*) are not apparent in the hemibrain dataset, though this may again be a limitation of the current annotation in which approximately half of the DN1ps are unaccounted for. However, strong DN1p to s-LNv connections are lacking even if we consider the three strong M cell targets, the SLP316s, to be missing DN1ps, as these form synapses on DN1ps and LPNs, but not the M group. Remarkably, there are no synaptic connections between the M and E1 groups, the latter of which was differentiated from E2 by the tight coupling of E1 molecular clocks to

those of the M group (*Yao and Shafer, 2014*). Thus, the strong coupling of M and E1 may take place through non-synaptic connections, consistent with recent work suggesting that the M group mediates its influence over endogenous timekeeping via non-synaptic signaling mechanisms, most likely via the non-synaptic release of PDF peptide (*Fernández et al., 2008*; *Yasuyama and Meinertzhagen, 2010*).

Compared to the other LN subgroups, the M group forms the smallest number of synaptic connections (*Figure 4C*). This group is also unique among the LN groups by virtue of the fact that all of its shared strong synaptic output targets form strong synaptic connections onto clock neurons (DN1ps and LPNs) (*Figure 6E*). Thus, the strongest synaptic outputs of the M groups appear to be intimately associated with the clock neuron network, either as neurons directly linking different clock neuron classes, or as currently unidentified clock neurons within the hemibrain dataset.

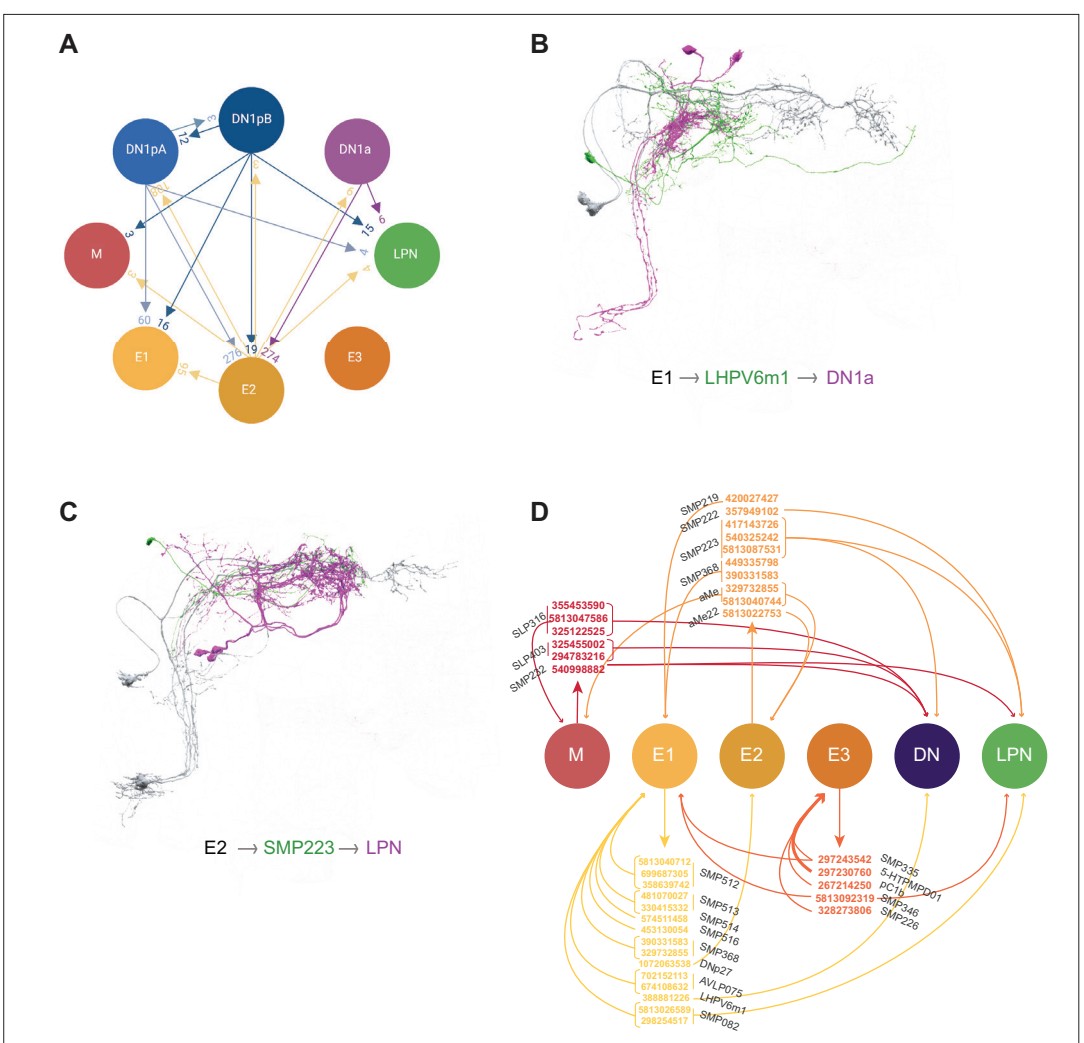

**Figure 10.** Connections within the clock neuron network. (**A**) Combined weights of medium and strong connections among different classes of identified clock neurons. Arrows indicate the direction of the connection. Some classes, such as E2, are strongly interconnected, while other clusters are relatively isolated. (**B**) Representative strong shared output of E1 that in turn contacts clock neurons. E1 are indicated in gray, their strong shared target LHPV6m1 cell 388881226 is indicated in green, and its strong target neurons DNa1 and 2 are indicated in magenta (the weight of output contacts is 19 and 10, respectively). (**C**) Representative strong shared output of E2 that in turn contacts clock neurons. E2 are indicated in gray, their strong shared target SMP223 cell 417143726 is indicated in green, and its strong target neurons LPN are indicated in magenta (the weight of output contacts is 14 to LPN, 18 to LPN2, and 20 to LPN3). (**D**) Summary of 'inter-clock' neurons providing strong poly synaptic connections between clock neurons.

## The E1 and E2 groups are major conduits of synaptic output from the clock network

Compared to the other three LN functional groups, the E1 group forms the largest number of connections onto postsynaptic neurons (*Figures 3F and 4C*) and the majority of their strong and shared synaptic targets do not themselves form synapses onto identified clock neurons within the hemibrain dataset. Of the minority of strong shared E1 targets that do synapse onto identified clock neurons, the majority form strong reciprocal connections with E1 itself (*Figures 7E and 10A*). Thus, the E group most strongly coupled with the critical M group (*Yao and Shafer, 2014*) also appears to be a major conduit of synaptic output from the clock neuron network.

Among the identified clock neurons within the hemibrain dataset, the E2 LNs (LNd6 and the 5th LNv) are clear outliers for synaptic connectivity with other clock neurons, forming strong synaptic connections onto E1 LNds and DN1ps. E2 LNds also receive strong synaptic inputs from DN1as and DN1ps and form strong connections with one another (*Figure 1C* and *Reinhard et al., 2022a*). This strong synaptic connectivity with other clock neurons may provide an explanation for the observation that the E2 group fails to synchronize its molecular clock with the M group when the clocks of the latter cells are slowed down, despite expressing the receptor for the M group's major circadian output peptide PDF (*Yao and Shafer, 2014*). Synaptic communication from non-M pacemakers may prevent E2 clocks from synchronizing with those of the M cells.

Like the strong shared synaptic targets of E1 LNds, the majority of E2 strong output targets do not synapse onto identified clock neurons. But, in contrast to E1, the minority of E2 targets that do form strong connections with identified clock neurons do so more broadly than clock recurrent E1 targets, forming strong synaptic connections onto the M group, E2, and the LPNs (*Figure 8E*). Though not as numerous as the synaptic partners of E1, E2 forms many more synaptic connections than either M or E3, and together E1 and E2 make up the bulk of synaptic output from the LN classes (*Figure 4C* and *Supplementary file 2*). Thus, there appear to be two major circadian output conduits from the LN clock neurons within the hemibrain volume, one of which is tightly coupled to the M group and the other whose output is likely more strongly shaped by other clock neuron classes (*Yao and Shafer, 2014*).

## The E3 group is characterized by a pattern of strong synaptic output that is distinct from E1 and E2 and is synaptically isolated from other identified clock neurons

The E3 LNds are distinct from the E1 and E2 groups, in that they form approximately half the number synaptic connections (*Figure 4C*). While clearly distinct from E1 and E2, the E3 group displays some similarity to the M group with respect to its connectivity to identified clock neurons in the hemibrain, forming very few connections with other clock classes while being interconnected by medium strength connections (*Figure 1C*). Among the identified clock neurons in the hemibrain, the strong output targets of E3 make strong reciprocal connections back to E3, but not to other identified clock neuron classes, though they do form a few medium strength connections onto LPNs (*Figure 9E*). This sets E3 clearly apart from the other LN groups, all of which have strong synaptic targets that form strong synapses onto other clock neuron classes (compare *Figures 6E–9E*). Thus, the E3 group appears to be uniquely isolated from other identified clock neuron classes for both direct and indirect synaptic connectivity.

The E3 group is unique among the LN clock neurons by virtue of its lacking CRY and PdfR expression (*Im et al., 2011*; *Yoshii et al., 2008*) and is therefore thought to be relatively isolated from both light/dark cycles (*Yoshii et al., 2010*) and PDF released from the LNvs. Indeed, the E3 group entrains its molecular clocks more readily to environmental temperature cycles than to light cycles (*Yoshii et al., 2010*) and does not synchronize with PDF expressing LNvs with slowed clocks (*Yao and Shafer, 2014*). A close examination of E3 output pathways and the extent to which they converge or fail to converge on endocrine, sleep, and premotor centers will offer important insights into how light and temperature are integrated by the circadian system to entrain to environmental cycles of both light and temperature, the latter of which lag behind the former in natural environments.

## Neurons without endogenous molecular clocks are likely integral to the central circadian pacemaker network

The LN clock neuron classes are critical nodes of circadian timekeeping and are predicted to communicate directly with endocrine, sleep, and premotor centers to drive circadian outputs (*Cavey et al., 2016*; *King and Sehgal, 2020*; *Potdar and Sheeba, 2018*; *Shafer and Keene, 2021*) and with other clock neuron classes to coordinate network timekeeping (reviewed by *Ahmad et al., 2021*). Thus, the LN classes are assumed to synapse upon other clock neurons to promote an endogenous sense of time and upon non-clock neurons to mediate daily changes in physiology and behavior. Our analysis reveals the presence of a third type of LN clock output target, neurons that mediate connections between identified clock neurons (e.g. *Figure 10B and C*).

All four functional classes of LN (M, E1, E2, and E3) provide strong synaptic outputs onto neurons that, in turn, form strong synaptic connections onto identified clock neurons within the hemibrain volume (*Figures 6E, F and 7E–9E*). In the case of E1 and E3 much of the strong output recurring to the clock neuron network is reciprocal. For example, the strong targets of E1 outputs preferentially form strong connections back onto E1 LNds (*Figure 7E*). For all four groups of LN, strong outputs recurring to the clock network form either strong or medium strength synapses on the LPNs, implicating this class as a particularly rich hub for polysynaptic inter-clock connectivity (*Figures 6E–10C*; *Reinhard et al., 2022a*).

Most of the neurons forming strong but indirect connections between identified clock neurons bear no resemblance to known clock neurons and are unlikely therefore to express the molecular circadian clock (e.g. *Figure 10B and C*). This finding suggests the need to expand our conception of what constitutes the fly's circadian clock network to include 'non-clock' neurons that provide strong synaptic connections between the neurons with endogenous molecular circadian clocks (*Figure 10D*). What role might such 'inter-clock-neurons' play within a circadian network? Networks underlying central pattern generators might offer clues.

Many central pattern generator networks consist of neurons both with and without endogenous pacemaking activity (reviewed by *Marder et al., 2005*). In such networks, connections between pacemaker neurons and follower neurons profoundly shape the time-course of rhythmic outputs, increase the precision of the central pattern generator, and provide a means for adjusting the phasing of its outputs (e.g. *Nadim et al., 2011*; *Purvis et al., 2006*; *Selverston and Miller, 1980*; *Weaver et al., 2010*). Given the distributed daily phases of neural activity displayed by the various classes of clock neurons (*Liang et al., 2016*), the circadian pace-making network represents a central pattern generator, albeit a lumberingly slow one. We hypothesize that the 'inter-clock neurons' we've identified here (e.g. *Figure 10B and C*) play significant roles in the determination of the circadian system's endogenous period, precision, and phasing of rhythmic outputs, much like the follower neurons of central pattern generators described above.

This testable hypothesis is just one of many that emerge from the comprehensive view of clock connectomics afforded by the hemibrain dataset (*Scheffer et al., 2020*). This remarkable picture of the synaptic connectivity displayed by clock neurons identified in this dataset (*Supplementary file 1* and *Supplementary file 2*), in conjunction with the open-source neuro-informatic resources available to the field (e.g. *Lazar et al., 2021*) and the highly specific genetic tools available for the manipulation of identified neurons (e.g. *Dionne et al., 2018*), will be the basis of a great deal of future work on the neural basis of circadian entrainment, timekeeping, and output.

## Methods

### Connectome data and neuron identification

All of the data analyzed in this study come from the Hemibrain v1.2.1 dataset made publicly available by Janelia Research Campus (*Scheffer et al., 2020*). The data details a full connectome derived from EM sections of a significant portion of the right hemisphere of a female *Drosophila* brain and small portions of the left hemisphere. These data were collected from a wild-type female brain reared under a 12 hr light, 12 hr dark cycle. The specimen was dissected at 5 days of age, 1.5 hr after lights-on (*Hulse et al., 2021*; *Scheffer et al., 2020*).

Hemibrain data was accessed via neuPrint and Fly Brain Observatory. Visualization of morphological data was done in the Fly Brain Observatory web interface (NeuroNLP.Hemibrain) while the

retrieval of connectome data was done with the Neuprint python package (https://github.com/connectome-neuprint/neuprint-python; *Berg et al., 2021*). Analyses were aided by the use of the pandas (https://pandas.pydata.org/), seaborn (https://seaborn.pydata.org/), and superVenn (https://github.com/gecko984/supervenn; *Indukaev, 2021*) python packages.

Each neuron in the database has a unique bodyID number and may also be identified with a cell-type name based on the name it was given in the literature or on its location and anatomy. Clock neurons were identified in the database from queries based on their known cell-type names. This means that there may be clock neurons that have been traced in the hemibrain, but if Janelia had not labeled them with their known cell name within the annotation, we have not included them as clock neurons here. The bodyID numbers for identified clock neurons were collected and used in the connectivity analyses to ensure consistency between platforms and future annotations.

The full table of clock neurons, their unique body IDs, and their sequential labels are provided below. The hemibrain annotation does not sequentially label instances of the same cell type. The Fly Brain Observatory does sequentially label cell type instances; however, there is no guarantee that the sequential labeling will consistently correspond to the specific body IDs. Thus, we employed a sequential labeling on a single retrieval from FBO and have used this labeling consistently throughout the paper (*Table 1*).

## Morphological data visualization with FlyBrainLab

FlyBrainLab (*Lazar et al., 2021*) is available as a python environment and a user-friendly web interface. The platform displays the hemibrain EM data and enables the analysis of the connectome data. The morphology images in our paper were generated in FlyBrainLab.

## Connectivity analysis

Synaptic connectivity data were retrieved from the Janelia hemibrain dataset (https://neuprint.janelia.org/). Our analyses included the synapses to or from traced but unnamed fragments as well as select orphan bodies, in addition to synapses from full, traced, and named neurons. Fragments are partial or truncated neurons that mainly lie beyond or at the boundaries of the hemibrain section. Although fragments were often unidentified cell types, we felt their inclusion was warranted to obtain as accurate a picture as possible of the relative amounts of synaptic connectivity to and from clock neurons.

In keeping with *Scheffer et al., 2020*, the synaptic strengths reported in our paper correspond to the number of postsynaptic densities counted on the postsynaptic neuron that are abutted by the presynaptic T-bar sites from the presynaptic neuron in question. Polyadic synapses, where a single presynaptic T-bar site contacts multiple postsynaptic sites, are common in the *Drosophila* brain (*Hulse et al., 2021*; *Scheffer et al., 2020*). Thus, synaptic weight for both the presynaptic and postsynaptic neuron is quantified as the postsynaptic density count.

We adopted the criteria from *Scheffer et al., 2020* to categorize connectivity strength. A weak connection is defined as a synaptic connection weight less than three. A medium strength connection is between three and nine synapses, while a strong connection is a strength equal to or greater than ten synapses. In our study, these criteria are applied to the total synapse count (i.e. the total number of postsynaptic densities) between two neurons rather than to the per-ROI synapse counts between them. Wherever applicable, the connection strength criteria used for any analysis or visualization is stated in the figure legend and in the main text.

The Sankey figures are based only on strong synaptic connections. Connectivity data were retrieved using the criteria for strong connections and exported into tables. The network structure of the clock connectome was graphed using the synaptic connectivity data retrieved from the hemibrain data and the networkX python package (https://networkx.org). Nodes are individual clock neurons and edges are labeled with the total synaptic connection weights from one neuron to another.

## Jaccard indices

The Jaccard similarity coefficient is defined as the ratio of the intersection and the union of two sets.

$$J\left(A, B\right) \; = \; \frac{A \cap B}{A \cup B}$$

In the context of our study, the Jaccard index represents the amount of overlap among the synaptic partners of two neurons. The Jaccard index is a number between 0 and 1, with 0 indicating no overlap and with 1 indicating that the sets of synaptic partners for two neurons are identical.

## Availability of analysis code

The data retrieval routines and the analyses used in this paper were done in the python coding language. We have made our scripts publicly available in a GitHub repository (*Li et al., 2022*): https://github.com/ClockConnectome/clock-connectome(copy archived at swh:1:rev:0c820068458b-d9249785c704716acfd1a9301f1c). The license can be found here: https://github.com/ClockConnectome/clock-connectome/blob/main/LICENSE.

## Acknowledgements

This work was supported by grants from the National Institute of Neurological Disorders and Stroke R01NS077933 to OTS and R01NS118012 to MPF and OTS, start-up funds from Barnard College (MPF) and the State of New York (OTS), grants NIH K22 NS104187, NSF NeuroNex Award DBI-1707398, and The Gatsby Charitable Foundation to GJG, grant NSF 2024607 to AAL and Barnard SRI funds to KL and AM We are grateful to Larry Abbot, Annika Barber, Aljoscha Nern, Michael Rosbash, and Gerald Rubin for helpful discussions and to Larry Abbot, Justin Blau, Abhilash Lakshman, Ashok Litwin-Kumar Paul Taghert, Aishwarya Ramakrishnan, and Robert Veline for kindly providing comments on the manuscript. *Figure 1A* was created in Biorender.

## Additional information

### Funding

| Funder | Grant reference number | Author |
|---|---|---|
| National Institute of Neurological Disorders and Stroke | R01NS118012 | Orie T Shafer Maria de la Paz Fernandez |
| National Institutes of Health | R01NS077933 | Orie T Shafer |
| National Institutes of Health | K22 NS104187 | Gabrielle J Gutierrez |
| National Science Foundation | NeuroNex Award DBI-1707398 | Gabrielle J Gutierrez |
| Gatsby Charitable Foundation | Research Award | Gabrielle J Gutierrez |
| National Science Foundation | Grant #2024607 | Aurel A Lazar |

The funders had no role in study design, data collection and interpretation, or the decision to submit the work for publication.

### Author contributions

Orie T Shafer, Conceptualization, Data curation, Investigation, Methodology, Writing - original draft, Writing - review and editing; Gabrielle J Gutierrez, Data curation, Formal analysis, Investigation, Methodology, Software, Supervision, Validation, Visualization; Kimberly Li, Formal analysis, Investigation, Software, Visualization; Amber Mildenhall, Formal analysis, Investigation, Visualization; Daphna Spira, Formal analysis, Investigation, Software; Jonathan Marty, Aurel A Lazar, Software, Visualization; Maria de la Paz Fernandez, Conceptualization, Formal analysis, Funding acquisition, Investigation, Project administration, Supervision, Visualization, Writing - original draft, Writing - review and editing

### Author ORCIDs

Orie T Shafer http://orcid.org/0000-0001-7177-743X
Gabrielle J Gutierrez http://orcid.org/0000-0002-2350-1559

Amber Mildenhall http://orcid.org/0000-0001-6495-8734
Aurel A Lazar http://orcid.org/0000-0003-4261-8709
Maria de la Paz Fernandez http://orcid.org/0000-0002-9261-6114

**Decision letter and Author response**
Decision letter https://doi.org/10.7554/eLife.79139.sa1
Author response https://doi.org/10.7554/eLife.79139.sa2

---

## Additional files

**Supplementary files**
• Supplementary file 1. Inputs onto clock neurons. List of Body IDs of all neurons that provide inputs onto currently identified circadian clock neurons and the weight of those pre-synaptic connections.

• Supplementary file 2. Outputs of clock neurons. List of Body IDs of all neurons that receive inputs from currently identified circadian clock neurons and the weight of those post-synaptic connections.

• MDAR checklist

**Data availability**
The current manuscript is a computational study, so no data have been generated for this manuscript. The dataset used was generated by Janelia Research Campus (*Drosophila* hemibrain connectome) and it is publicly available: https://neuprint.janelia.org/ The original manuscript (Scheffer et al., 2020) can be found here: https://doi.org/10.7554/eLife.57443.

The following previously published dataset was used:

| Author(s) | Year | Dataset title | Dataset URL | Database and Identifier |
|-----------|------|---------------|-------------|--------------------------|
| Li PH, Lindsey L, Neubarth N, Olbris DJ, Otsuna H, Trautman ET, Ito M, Bates AS, Goldammer J, Wolff T, Svirskas R, Schlegel P, Neace E, Knecht CJ, Alvarado CX, Bailey DA, Ballinger S, Borycz JA, Canino BS, Cheatham N, Cook M, Dreher M, Duclos O, Eubanks B, Fairbanks K, Finley S, Forknall N, Francis A, Hopkins GP, Joyce EM, Kim S, Kirk NA, Kovalyak J, Lauchie SA, Lohff A, Maldonado C, Manley EA, McLin S, Mooney C, Ndama M, Ogundeyi O, Okeoma N, Ordish C, Padilla N, Patrick CM, Paterson T, Phillips EE, Phillips EM, Rampally N, Ribeiro C, Robertson MK, Rymer JT, Ryan SM, Sammons M, Scott AK, Scott AL, Shinomiya Aya, Smith C, Smith K, Smith NL, Sobeski MA, Suleiman A, Swift J, Takemura S, Talebi I, Tarnogorska D, Tenshaw E, Tokhi T, Walsh JJ, Yang T, Horne JA, Li F, Parekh R, Rivlin PK, Jayaraman V, Costa M, Jefferis GSXE, Ito Kei, Saalfeld S, George R, Meinertzhagen IA, Rubin GM, Hess HF, Jain V, Stephen MP, Louis KS, Shan CX, Michal J, Zhiyuan L, Shin-ya T, Kenneth JH, Gary BH, Kazunori S, Jeremy MS, Stuart B, Jody C, Philip MH, William TK, Lowell U, Ting Z, David A, Tim B, John B, Tom D, Dagmar K, Takashi K, Khaled AK, Laramie L | 2020 | Resource Collection for a Connectome and Analysis of the Adult *Drosophila* Central Brain | https://neuprint.janelia.org/ | Hemibrain dataset v1.2.1, v1.2.1 |

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
