## [Editor Report]

This paper analyzes the synaptic connections of two subsets of the *Drosophila* clock neurons, the small ventral lateral neurons, and the dorsal lateral neurons. The study reveals high heterogeneity of the dorsal lateral neurons and the potential involvement of non-clock neurons that might act as inter-clock neurons. The manuscript will be of interest to chronobiologists and neuroscientists working on neuronal networks and it provides new insights into circadian clock network organization that may be of general value. The data analysis is rigorous, and the conclusions are justified by the data.

---

## [Decision Letter]

**Decision letter after peer review:**

Thank you for submitting your article "Connectomic Analysis of the *Drosophila* Lateral Neuron Clock Cells Reveals the Synaptic Basis of Functional Pacemaker Classes" for consideration by *eLife*. Your article has been reviewed by 3 peer reviewers, and the evaluation has been overseen by Claude Desplan as the Senior and Reviewing Editor. The following individual involved in the review of your submission has agreed to reveal their identity: Charlotte Helfrich-Förster (Reviewer #1).

Essential revisions:

The paper is already very dense and we do not want to add more work to it, although one reviewer would have liked to have some of the ideas that are proposed be tested. We ask you to take into consideration all the editorial suggestions from reviewer #1 and Reviewer #2 and make sure to help the reader with a complex paper.

The list of comments is long and extensive but will improve the paper significantly and should be feasible in a few weeks without further experiments.

*Reviewer #1 (Recommendations for the authors):*

Although it is impossible to avoid abbreviations, these should be reduced to a minimum. For example, all words that are used less than 5 times should not be abbreviated (e.g., CPG). I also recommend talking about CRY-positive or CRY-negative clock neurons instead of CRY+ and CRY- clock neurons (the same applies for PDF-positive and -negative neurons).

In many figures, the lettering is too small, and/or gray letters are used instead of black ones. This makes reading very difficult. Similarly, many drawings of individual neurons are too small, and lines are too faint or too thin. This needs general improvement. Below, I will only mention the most evident cases.

Abstract lines 27-30: "Moreover, we identify a small number of clock cell subtypes representing highly synaptically coupled nodes within the clock neuron network. This suggests that neurons lacking molecular timekeeping likely play integral roles within the circadian timekeeping network." I do not see why the small number of clock cell subtypes representing synaptically coupled nodes should suggest that neurons lacking molecular timekeeping play integral roles within the clock network. Something is missing here.

Line 110: The author may wish to also cite the paper of Reinhard et al. (2022b) (https://doi.org/10.3389/fphys.2022.886432) that describes the dorsal clock neurons in more detail.

Line 117: the lettering in Figure 1C is too small so that it is impossible to find the LNd6.

Line 119: This is the wrong citation. Please cite the paper mentioned above.

Line 142: The 5th LNv was first described by Kaneko et al. (1997, JN 17:6745-6760). Perhaps it is fair to cite this paper here, too.

Lines 152-154: Schubert et al. did show that the cell body of the 5th s-LNv is significantly larger than those of the s-LNvs (it has exactly the same size as that of the LNd6). Furthermore, its soma is located more dorsally than that of the s-LNvs. Therefore, Schubert et al. suggest calling this cell not anymore 5th s-LNv, because it is not an s-LNv. Indeed, subsequent papers from this lab just called it 5th LN. Please think about changing the name of the cell to 5th LN.

Line 197: "connectomic" should be replaced by "transcriptomic".

Page 9 and 10: I find the description of the different LNds very puzzling and partly redundant. It may be helpful to enlarge Figure 3D (perhaps even as an extra Figure) and not only include the synaptic connections between the 6 LNds in this figure but also add their neuropeptide/transmitter expression. In the text, it might help to combine the description of neuropeptide expression (lines 240-243) with neuropeptide/neurotransmitter expression (lines 253-258). Furthermore, it might help to mention PDF-receptor expression already earlier together with CRY-expression (lines 231-233).

I also wonder, why ITP as neuropeptide of the 5th LN is only mentioned here (line 243) not already earlier when the 5th LN was initially described.

Figure 3C: The green dots representing inter LNd contacts cannot be seen, even not when amplified to 200%. The same applies to several other figures.

Figure 3D: The labeling of the LNd1-6 needs to be enlarged. Furthermore, the white labeling on the gray background is hard to see. Why do their names carry an additional "R"?

Line 267: The second neuropeptide of the s-LNv should already be mentioned at the place, where these neurons were initially described (e.g. page 5). Furthermore, their fast neurotransmitter expression (Frenkel et al., 2017) should be mentioned there, too. This citation is currently completely lacking.

Line 431: I miss a clear statement that neuropeptidergic signaling could represent an essential output from the clock network to descending neurons. This is only briefly mentioned in lines 501-504 with respect to the neuropeptide PDF. Since all clock neurons appear to be peptidergic and many clock neurons express even more than one neuropeptide, I suggest mentioning in the first paragraph of the discussion that not only electrical synapses likely contribute to circadian timekeeping but also neuromodulation through neuropeptides.

Line 401-403: This sentence does not describe what the figure shows if I understand it correctly. In Figure 9E, the connectivity of the postsynaptic E3 neurons with other clock neurons is shown, not the connectivity of the E3 with other clock neurons.

Line 433: Please mention the timepoint at which the hemibrain was generated also here and not only in M&M.

Line 463: Please cite here additionally the paper of Schubert et al., because these authors were the first who showed the morphological similarity between these two neurons.

Line 527: This statement might be supported by the study of Reinhard et al. (2022b) who found the same.

*Reviewer #2 (Recommendations for the authors):*

I think it is a very nice piece of work that clearly deserves publication and I only have minor comments.

1. Some of the clock neurons, including LNds, send axons in the dorsal brain that go through the midline and end in the other hemisphere (see papers from C Helfrich-Forster). As indicated in the text, the Janelia farm dataset does not include this information about contralateral connections since it contains one hemisphere only. However, this important feature of some clock neurons is not mentioned in the manuscript and I think that the information that is available in the literature should be discussed when using axonal projections to define the different neuronal subsets.

2. Page 7: "and all four receive strong or medium strength synaptic inputs from these three presynaptic cell types (Figure 4D; Supplemental Table S1)."

Figure 4D shows only two presynaptic cells, please clarify.

3. Page8: "Among the inputs targeting only one s-LNv, and mediated by only three synapses, is the HB-eyelet, a surprising finding given the long-held model that this external photoreceptor provides direct excitatory input onto s-LNvs and contributes to light entrainment of circadian rhythms".

The long-held model was already challenged by the finding that the absence of histamine transmission induces a loss of circadian entrainment (Alejevski et al.,2019).

4. In Figures 2D, and E (inputs and outputs), please add the name of the non-LNv clock cells for the corresponding ID numbers.

5. In figure 3E, F: please indicate for each LNd the number of synapses with strong partners as in figure 2D, E for the LNvs.

*Reviewer #3 (Recommendations for the authors):*

In this manuscript, the authors used a publicly available dataset to extract information about connectivity in the circadian network. I found this paper hard to evaluate. It does not rely on newly obtained data, only on the analysis of existing information. I am not an expert in EM data analysis and do not know if the analysis employed here is particularly sophisticated. Even relatively simple analysis could in principle be OK if it leads to significant biological insight. In this case, there are some ideas that are proposed but are not tested, so it is difficult to know what the value of the paper will be in the future. I could see it being relevant for people in the circadian field, but the authors should convince us by experimentally testing at least one of the predictions generated by their analysis.

---

## [Author Response]

Reviewer #1 (Recommendations for the authors):Although it is impossible to avoid abbreviations, these should be reduced to a minimum. For example, all words that are used less than 5 times should not be abbreviated (e.g., CPG). I also recommend talking about CRY-positive or CRY-negative clock neurons instead of CRY+ and CRY- clock neurons (the same applies for PDF-positive and -negative neurons).

We agree with the reviewer that these changes will make the text easier to read. “CPG” has been replaced with “central pattern generators” and “aMe” has been replaced with “accessory medulla”. All abbreviations that remain are used five times or more in the revised manuscript (SCN, PDF, etc.). “Cry+” and “Cry” have been replaced with “Cry-positive” and “Cry-negative”, respectively, throughout the manuscript.

In many figures, the lettering is too small, and/or gray letters are used instead of black ones. This makes reading very difficult. Similarly, many drawings of individual neurons are too small, and lines are too faint or too thin. This needs general improvement. Below, I will only mention the most evident cases.

The font size has been increased in all the figures, and changed from gray to black. The sizes of all images of individual neurons have increased, and we have improved the resolution when needed. There are times when the number of cell IDs are too numerous to make all the text in a figure significantly larger (like figure 3F). However, in these cases the major result in the figure is not dependent on the exact identification of each synaptic partner, but rather the number of synaptic partners shared by specific clock neurons. The exact identification of each synaptic partner can also be found in the supplementary tables (inputs and outputs).

Abstract lines 27-30: "Moreover, we identify a small number of clock cell subtypes representing highly synaptically coupled nodes within the clock neuron network. This suggests that neurons lacking molecular timekeeping likely play integral roles within the circadian timekeeping network." I do not see why the small number of clock cell subtypes representing synaptically coupled nodes should suggest that neurons lacking molecular timekeeping play integral roles within the clock network. Something is missing here.

The reviewer is correct, this was a typo. It should have said “non-clock”, and we have modified it in the abstract.

Line 110: The author may wish to also cite the paper of Reinhard et al. (2022b) (https://doi.org/10.3389/fphys.2022.886432) that describes the dorsal clock neurons in more detail.

This reference has been added (lines 128-129).

Line 117: the lettering in Figure 1C is too small so that it is impossible to find the LNd6.

This has been corrected in Figure 1C and in most other figures.

Line 119: This is the wrong citation. Please cite the paper mentioned above.

This citation has been replaced with Reinhard et al. (2022b).

Line 142: The 5th LNv was first described by Kaneko et al. (1997, JN 17:6745-6760). Perhaps it is fair to cite this paper here, too.

This citation has been added (line 158).

Lines 152-154: Schubert et al. did show that the cell body of the 5th s-LNv is significantly larger than those of the s-LNvs (it has exactly the same size as that of the LNd6). Furthermore, its soma is located more dorsally than that of the s-LNvs. Therefore, Schubert et al. suggest calling this cell not anymore 5th s-LNv, because it is not an s-LNv. Indeed, subsequent papers from this lab just called it 5th LN. Please think about changing the name of the cell to 5th LN.

We now introduce the “5th s-LNv” historically and then cite Schubert et al.’s re-naming of this neuron as the “5th LNv.” We refer to these neurons by this name for the remainder of the study. The figures and figure legends have also been modified to reflect this change.

Line 197: "connectomic" should be replaced by "transcriptomic".

In this case, we were referring to recent work from the Sprecher lab on the larval connectome, whose analysis included larval LNvs. We have clarified this by revising the sentence in question as follows:

“A recent connectomic analysis of the larval brain by Larderet and colleagues reached similar conclusions regarding the uniformity of sensory inputs, anatomy, and local connections among the four larval *pdf*+ neurons (Larderet et al., 2017), which persist through metamorphosis to become the adult s-LNvs (Helfrich-Förster, 1997).”

Page 9 and 10: I find the description of the different LNds very puzzling and partly redundant. It may be helpful to enlarge Figure 3D (perhaps even as an extra Figure) and not only include the synaptic connections between the 6 LNds in this figure but also add their neuropeptide/transmitter expression. In the text, it might help to combine the description of neuropeptide expression (lines 240-243) with neuropeptide/neurotransmitter expression (lines 253-258). Furthermore, it might help to mention PDF-receptor expression already earlier together with CRY-expression (lines 231-233).I also wonder, why ITP as neuropeptide of the 5th LN is only mentioned here (line 243) not already earlier when the 5th LN was initially described.

We agree that our summary of the anatomical subclasses was redundant and not as clear as it could have been. We have edited this section of the manuscript to avoid redundancy and have revised Figure 3D to make the anatomical and functional subclasses of the LNds clearer. For the sake of consistency, we have also revised Figure 2D and now show not only synaptic connections between the 4 sLNvs but also their neuropeptide expression. We have also, as suggested, described PdfR expression earlier in the text, and now introduce it with our initial description of Cry expression. In addition, we have mentioned that the 5th LNv expresses both ITP and CHAT when we first introduce these neurons in the section introducing the LNv classes.

Figure 3C: The green dots representing inter LNd contacts cannot be seen, even not when amplified to 200%. The same applies to several other figures.

We have replaced the images in 2C and 3C and now use orange instead of green for inter-group contacts, which makes them more visible. We have also substantially increased the size of panels 2A-C, 2E-F, and all other panels with neuron morphology in the main figures.

Figure 3D: The labeling of the LNd1-6 needs to be enlarged. Furthermore, the white labeling on the gray background is hard to see. Why do their names carry an additional "R"?

Our new panel 3D has larger labeling and does not include the “R”. This was a typo, left-over from the NeuPrint nomenclature for the individual annotated neurons.

Line 267: The second neuropeptide of the s-LNv should already be mentioned at the place, where these neurons were initially described (e.g. page 5). Furthermore, their fast neurotransmitter expression (Frenkel et al., 2017) should be mentioned there, too. This citation is currently completely lacking.

We agree. We now describe sNPF and glycine expression by the sLNvs earlier in the Results section. We have also incorporated the *Frenkel et al., 2017* reference (lines 148-150).

Line 431: I miss a clear statement that neuropeptidergic signaling could represent an essential output from the clock network to descending neurons. This is only briefly mentioned in lines 501-504 with respect to the neuropeptide PDF. Since all clock neurons appear to be peptidergic and many clock neurons express even more than one neuropeptide, I suggest mentioning in the first paragraph of the discussion that not only electrical synapses likely contribute to circadian timekeeping but also neuromodulation through neuropeptides.

We agree that such paracrine/peptide connections are likely important for timekeeping and should be mentioned when discussing the limitations of connectomics focused on chemical synapses. We have added two sentences to the first paragraph of the discussion, as suggested:

“In addition, neuropeptides can be released non-synaptically (e.g. (Yasuyama & Meinertzhagen, 2010)), act over long distances, and such paracrine signaling likely contributes to circadian timekeeping in the brain (e.g. (Maywood, Chesham, O'Brien, & Hastings, 2011)). The connectomic analysis presented here does not include such peptidergic connectivity.”

Line 401-403: This sentence does not describe what the figure shows if I understand it correctly. In Figure 9E, the connectivity of the postsynaptic E3 neurons with other clock neurons is shown, not the connectivity of the E3 with other clock neurons.

The reviewer is correct. We have clarified this (lines 429-431), and this section now reads:

“With regard to its shared strong outputs, the E3 subgroup forms strong shared synaptic connections onto only 16 neurons (Figure 3F and 4G), five of which form synapses onto identified clock neurons within the hemibrain (Figure 9E). The synapses that those five neurons form back onto clock neurons are limited to LNds and LPNs, with most connections forming back onto E3 LNds themselves (Figure 9A-C and E).”

Line 433: Please mention the timepoint at which the hemibrain was generated also here and not only in M&M.

This information has been incorporated (line 471).

Line 463: Please cite here additionally the paper of Schubert et al., because these authors were the first who showed the morphological similarity between these two neurons.

This reference has been added (line 506).

Line 527: This statement might be supported by the study of Reinhard et al. (2022b) who found the same.

This reference has been added (line 640).

Reviewer #2 (Recommendations for the authors):I think it is a very nice piece of work that clearly deserves publication and I only have minor comments.1. Some of the clock neurons, including LNds, send axons in the dorsal brain that go through the midline and end in the other hemisphere (see papers from C Helfrich-Forster). As indicated in the text, the Janelia farm dataset does not include this information about contralateral connections since it contains one hemisphere only. However, this important feature of some clock neurons is not mentioned in the manuscript and I think that the information that is available in the literature should be discussed when using axonal projections to define the different neuronal subsets.

We agree that this is an important consideration for our study and that we could have said more about which of the annotated clock neuron classes within the hemibrain are known to project contralaterally. We have therefore added the following to the opening paragraph of the discussion:

“Several classes of clock neurons, including the large LNvs, the 5th LNv, the LNds, and DN1ps, project across the midline and form connections in the contralateral hemisphere (Helfrich-Forster, Yoshii, Wulbeck, Grieshaber, Rieger, Bachleitner, Cusumano, et al., 2007). Many of these connections are expected to reside outside of the hemibrain volume. Our estimates of inter-clock connectivity will therefore not include these contralateral connections.”

2. Page 7: "and all four receive strong or medium strength synaptic inputs from these three presynaptic cell types (Figure 4D; Supplemental Table S1)."Figure 4D shows only two presynaptic cells, please clarify.

We have clarified this point (lines 198-200):

“Only three neurons in the hemibrain provide ten or more synapses onto at least one of these cells (Figure 2D), and only two of those three neurons form strong connections onto all four sLNvs (Figure 4D; Supplemental Table S1).”

3. Page8: "Among the inputs targeting only one s-LNv, and mediated by only three synapses, is the HB-eyelet, a surprising finding given the long-held model that this external photoreceptor provides direct excitatory input onto s-LNvs and contributes to light entrainment of circadian rhythms".The long-held model was already challenged by the finding that the absence of histamine transmission induces a loss of circadian entrainment (Alejevski et al.,2019).

We have added the following sentence and reference (lines 219-221):

The sparse connectivity between the eyelet and the s-LNVs found within the hemibrain is consistent with recent work suggesting that the eyelet makes negligible contributions to light entrainment (Alejevski et al., 2019).

4. In Figures 2D, and E (inputs and outputs), please add the name of the non-LNv clock cells for the corresponding ID numbers.

We have added this information to Figure 2, and also to Figure 3 for consistency.

5. In figure 3E, F: please indicate for each LNd the number of synapses with strong partners as in figure 2D, E for the LNvs.

We have incorporated this information to Figure 3E and F.